# Controllable Satellite-to-Street-View Synthesis with Precise Pose Alignment and Zero-Shot Environmental Control

**Xianghui Ze[1]**[*], **Zhenbo Song[1]**, **Qiwei Wang[2]**, **Jianfeng Lu[1]**, **Yujiao Shi[2]**[†]

[1]Nanjing University of Science and Technology, [2]ShanghaiTech University
```
{zexh, songzb, lujf}@njust.edu.cn,
{wangqw2024, shiyj2}@shanghaitech.edu.cn
```

## Abstract

Generating street-view images from satellite imagery is a challenging task, particularly in maintaining accurate pose alignment and incorporating diverse environmental conditions. While diffusion models have shown promise in generative tasks, their ability to maintain strict pose alignment throughout the diffusion process is limited. In this paper, we propose a novel Iterative Homography Adjustment (IHA) scheme applied during the denoising process, which effectively addresses pose misalignment and ensures spatial consistency in the generated street-view images. Additionally, currently, available datasets for satellite-to-street-view generation are limited in their diversity of illumination and weather conditions, thereby restricting the generalizability of the generated outputs. To mitigate this, we introduce a text-guided illumination and weather-controlled sampling strategy that enables fine-grained control over the environmental factors. Extensive quantitative and qualitative evaluations demonstrate that our approach significantly improves pose accuracy and enhances the diversity and realism of generated street-view images, setting a new benchmark for satellite-to-street-view generation tasks.

## 1 Introduction

This paper tackles the problem of satellite-to-street-view synthesis, aiming to generate street-view images that are geometrically consistent with satellite imagery under a pre-determined relative pose and diverse environmental conditions. Synthesizing such images has critical applications in urban modeling, geospatial analysis, and virtual reality. While satellite images provide comprehensive global coverage at low cost, capturing ground-level data is resource-intensive and lacks access to varied weather and seasonal conditions. Therefore, generating geometrically accurate street-view images with controllable environmental diversity is a challenging but essential task.

The difficulties in this task stem from two key aspects: geometric consistency and environmental diversity. Geometric consistency is challenging due to the vast perspective difference between satellite and ground views where satellite images provide a top-down perspective, street-view images show the scene from a lateral angle, resulting in minimal visual overlap. Establishing a reliable mapping between these two viewpoints requires precise handling of geometric information. Additionally, environmental diversity is crucial for practical applications, as street-view images under different weather and lighting conditions are needed to simulate real-world scenarios. Existing datasets for satellite-to-street-view tasks offer limited variation in environmental conditions, further complicating the generation of diverse and realistic outputs.

Recent works have made progress in this domain by using geometric priors to bridge the gap between satellite and ground perspectives. For example, Shi et al. (2022a) used multi-plane images to infer depth maps from satellite data and generate ground-level images, while Qian et al. (2023) rendered panoramas from satellite images based on density fields. However, these approaches are prone to errors due to their reliance on approximate 3D priors. Meanwhile, methods focusing on generating images with environmental diversity (Assion et al. (2024)) are often limited to in-domain editing and lack the ability to generate diverse scenes from satellite inputs.

---

[*]This work was done during a visit at ShanghaiTech University.
[†]Corresponding author.

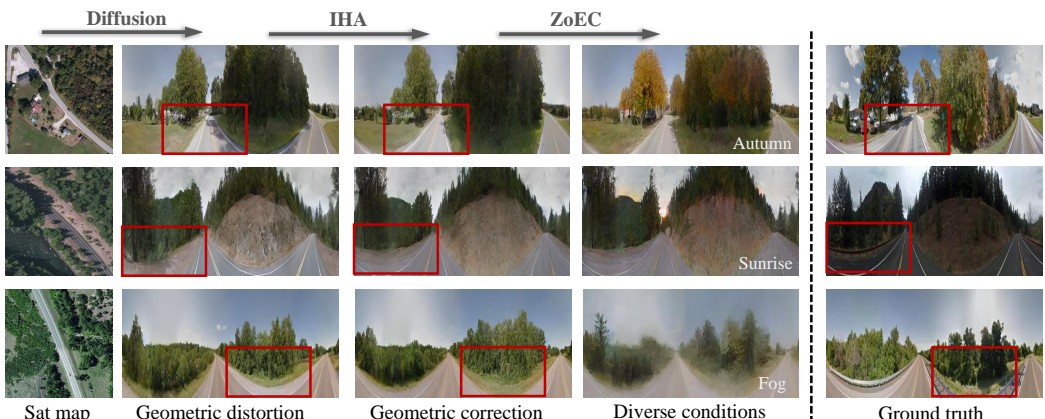

Figure 1: Our method synthesizes ground-level images from satellite maps by integrating Iterative Homography Adjustment (IHA) to refine geometric alignment and Text-guided Zero-shot Environmental Control (ZoEC) to provide flexible environmental control, enabling precise pose alignment and diverse scene generation.

As shown in Fig. 1, this paper proposes a novel framework for satellite-to-street-view synthesis that addresses two key challenges: ensuring geometric alignment and enabling environmental control. Our approach is based on the classical stable diffusion model (Rombach et al. (2022b)), which has shown strong performance in image synthesis but faces significant challenges in maintaining precise pose information during generation, often leading to misalignment between generated street-view images and satellite inputs.

To overcome these issues, we propose two key innovations. First, unlike standard diffusion models where conditions are implicitly encoded within the denoising U-Net, we introduce a cross-view conditioning mechanism that incorporates geometric information. This mechanism ensures that the generated street-view panoramas maintain a consistent spatial layout, aligning scene objects with the satellite images. Second, we propose an Iterative Homography Adjustment scheme that operates during the diffusion sampling phase. This process corrects pose misalignment by iteratively adjusting the intermediate output based on the relative pose difference between the generated image and the satellite view, ensuring geometric consistency throughout the generation process.

Furthermore, real-world environments exhibit a wide range of illumination and weather conditions, which are rarely reflected in existing satellite-and-street-view datasets. This lack of diversity limits the model's ability to generate realistic street-view images under different environmental settings. To overcome this limitation, we introduce a Zero-Shot Environmental Control strategy, which uses text prompts to guide the generation of street-view images under varying illumination and weather conditions without retraining the model.

Finally, since satellite images lack crucial details like the sky and the sides of buildings that are present in ground-level views, traditional metrics such as RMSE are inadequate for fair comparisons across different methods. To provide a more comprehensive evaluation, we introduce new evaluation metrics that assess both semantic and geometric consistency between generated and ground truth images, as well as the alignment between the generated images, pose, and environmental conditions.

Our contributions are summarized as follows:

- We introduce a novel cross-view conditioning mechanism that incorporates geometric information, ensuring precise spatial alignment between the generated street-view panoramas and satellite images.
- We propose an Iterative Homography Adjustment scheme during the diffusion sampling process, addressing pose misalignment by iteratively correcting the generated images for geometric consistency.
- We demonstrate the effectiveness of our framework in generating diverse street-view images with controllable environmental conditions (e.g., weather and lighting) in a zero-shot manner.
- We design new evaluation metrics that measure both semantic and geometric consistency between the generated and ground truth images, as well as the alignment between the generated outputs and the controllable pose and environmental conditions.

## 2 RELATED WORK

**Cross-view ground scene generation.** In the study by Zhai et al. (2017), the authors first attempted to align the semantic features of satellite images onto ground-level perspectives. In Wu et al. (2022), GANs were employed to generate ground images. Strong geometric relationships were introduced in the task of ground image generation by Lu et al. (2020); Shi et al. (2022a); Qian et al. (2023). Li et al. (2021; 2024b) explicitly constructed a 3D point cloud representation of the scene, and then transformed it into a scene representation in a feed-forward manner. Gao et al. (2024a); Xu & Qin (2024); Li et al. (2024a) advocate for generating ground images from ground-to-ground scene segmentation images. Among these, Gao et al. (2024a) specifically highlights the impact of various noises on the generated results and innovatively proposes a noise-prior-based solution. However, previous methods have predominantly relied on coarse scene priors, leading to compounded errors in the results. We propose GCA and IHA to ensure geometric consistency between ground images and satellite views. The cross-view generation work targeting single objects is also highly inspiring. Liu et al. (2023) overlays camera position encoding for scene transformation, while Poole et al. (2022) utilizes diffusion to optimize the Nerf representation of scenes. Melas-Kyriazi et al. (2024) and Gao et al. (2024b) generate continuous frame data based on video diffusion. These approaches often fail in large-scale scene reconstruction, especially when dealing with significant perspective differences between satellite and ground images, which is the issue we are dedicated to addressing.

**Text-controlled image generation.** In text generation, a multitude of solutions have emerged over time leveraging Generative Adversarial Networks (GANs) (Dash et al. (2017); Regmi & Borji (2018); Ruan et al. (2021); Tao et al. (2022)). However, with the introduction of diffusion (Song et al. (2020); Rombach et al. (2022b)), its exceptional generation capability has evolved into a potent tool for creating images. Significant strides have been taken in text-driven image synthesis through diffusion by Avrahami et al. (2022); Li et al. (2023b); Brooks et al. (2023). Li et al. (2023a); Gao et al. (2024c) propose a method that generates ground images based on text conditions and BEV segmentation images. However, this strategy is hampered by the limitations of expressive capabilities in scene segmentation, leading to arbitrary results in scene synthesis. In this paper, we employ satellite images with enhanced representational capabilities for ground synthesis and introduce a novel text-guided mechanism to ensure both the reliability of scene generation and the diversity of generated results.

## 3 METHOD OVERVIEW

Our task is to generate realistic street-view images based on a given satellite image, a specified relative pose, and an environmental factor such as illumination or weather. We use a latent diffusion model framework to conditionally synthesize these street-view images while ensuring control over both geometric and environmental aspects. The geometric condition is encoded via the satellite image and relative pose, while environmental control is applied during inference to achieve diverse visual effects.

### 3.1 DIFFUSION MODEL FRAMEWORK

To generate street-view images with controlled conditions, we leverage a latent diffusion model (Rombach et al. (2022a)). Let the target street-view image be denoted as $x_0$ and its corresponding latent embedding as $z_0$. Using an image decoder $\mathcal{D}(\cdot)$, the target image can be reconstructed as $x_0 = \mathcal{D}(z_0)$. The forward diffusion process adds Gaussian noise to the latent embedding $z_0$ progressively, resulting in $z_t$ at each time step $t \in [0, T]$, where the noise level is defined by $\beta_t \in (0, 1)$:

$$z_t = \sqrt{\bar{\alpha}_t} z_0 + \sqrt{1 - \bar{\alpha}_t} \varepsilon \tag{1}$$

where $\varepsilon \sim \mathcal{N}(0, I)$, $\alpha_t = 1 - \beta_t$, and $\bar{\alpha}_t = \prod_{i=0}^{t} \alpha_i$. The denoising U-Net, $\epsilon_\theta(z_t, t, c)$, is used to predict the added noise, with condition $c$ representing the posed embedding of the satellite image.

The reverse diffusion process is based on DDIM sampling:

$$p(z_{t-1}|z_t) = \mathcal{N}(z_{t-1}; \mu_\theta(z_t, t, c), \sigma_t^2 I) \tag{2}$$

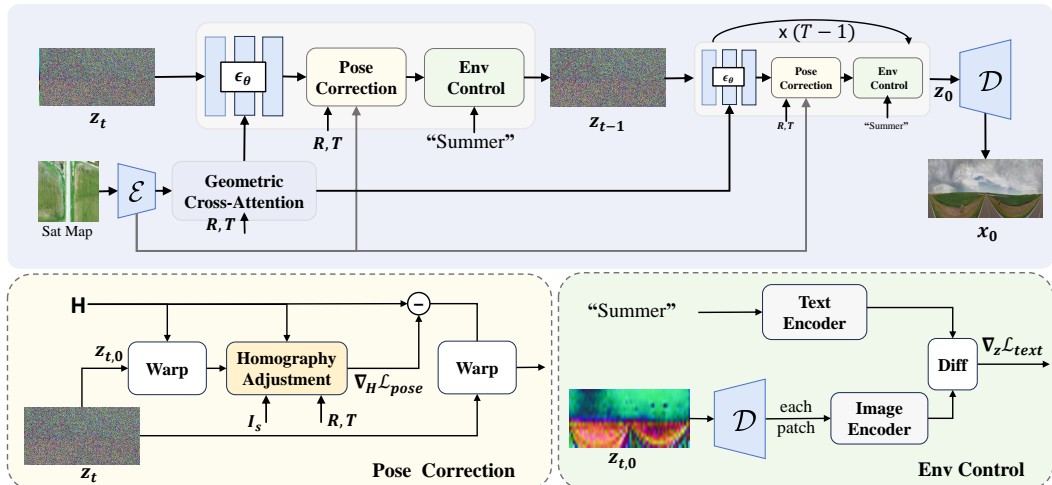

Figure 2: Overview of our framework. Our approach aims to utilize satellite images to generate corresponding ground images. We leverage geometric relationships extensively and have the capability to alter the features of the generated images based on different text prompts.

where $\sigma_t$ controls the sampling's stochasticity, and $\mu_\theta(z_t, t, c)$ is calculated as:

$$\mu_\theta(z_t, t, c) = \sqrt{\bar{\alpha}_{t-1}} \left( \frac{z_t - \sqrt{1 - \bar{\alpha}_t} \cdot \epsilon_\theta(z_t, t, c)}{\sqrt{\bar{\alpha}_t}} \right) + \sqrt{1 - \bar{\alpha}_{t-1} - \sigma_t^2} \epsilon_\theta(z_t, t, c) + \sigma_t \varepsilon \quad (3)$$

## 3.2 ENVIRONMENTAL GUIDANCE AND POSE ALIGNMENT

Due to the lack of labeled environmental conditions (e.g., variations in weather and lighting) in the existing cross-view image datasets, using classifier-free guidance for the control, like satellite image features and the relative pose, during the training process is impossible. To address this issue, we propose a zero-shot environmental guidance approach using text prompts by leveraging a classifier guidance strategy. Specifically, we employ off-the-shelf CLIP text embeddings as environmental control, enabling the guidance of environmental factors like lighting and weather during inference. Furthermore, implicitly leveraging satellite image features as a condition often leads to pose misalignment in generated street-view images, as demonstrated in Fig.6. To further enhance the pose alignment, we introduce an Iterative Homography Adjustment within the DDIM sampling process. Specifically, we denote environmental and pose guidance as $g_{\text{text}}$ and $g_{\text{pos}}$, respectively. As shown in Appendix A.4, Using classifier-guidance (Dhariwal & Nichol (2021)), the denoising process with environmental and pose guidance is defined as:

$$z_{t-1} = \mu_\theta(z_t, t, c) + \lambda \nabla_{z_t} \log p(g_{\text{pose}}|z_t) + \gamma \nabla_{z_t} \log p(g_{\text{text}}|z_t) + \sigma_t \varepsilon \quad (4)$$

where hyperparameters $\gamma$ and $\lambda$ control the strength of environmental and pose conditioning, respectively, ensuring that the generated image meets both environmental and geometric requirements.

In the following section, we discuss how conditions $c$ of the posed satellite image are encoded, along with further technical details on enforcing pose alignment through $\nabla_{z_t} \log(p(g_{\text{pose}}|z_t))$ and applying environmental guidance via $\nabla_{z_t} \log(p(g_{\text{text}}|z_t))$ during the inference process.

## 4 THE PROPOSED FRAMEWORK

This paper aims to generate ground-level images that accurately align with a given satellite image, a relative pose, and environmental conditions. To achieve precise pose control, as illustrated in Fig. 2, we propose embedding satellite features and the relative pose information into the diffusion model through two complementary mechanisms: a Geometric Cross-Attention module within the denoising network and an iterative pose enhancement strategy during the inference denoising stage. Considering diverse weather and illumination data are absent in the training set, we design a Zero-Shot Environmental Control strategy that allows flexible control over scene variations during inference without requiring additional training data.

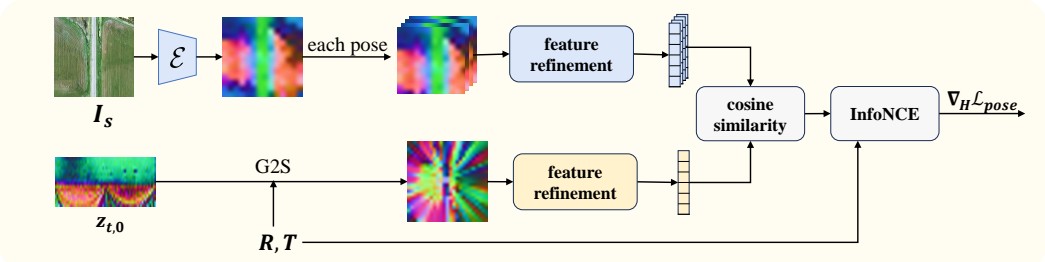

Figure 3: Homography Adjustment. We compare ground-level images with satellite images, calculating the loss by assessing the relative pose of the ground images at time t against a predefined pose. Ground images are then adjusted based on this comparison.

## 4.1 CROSS-VIEW CONDITIONING MECHANISM

Instead of implicitly encoding global feature vectors of posed satellite images, we propose a Geometric Cross-Attention (GCA) mechanism that explicitly constructs geometric correspondences.

Given a pixel coordinate $(u_g, v_g)$ in the target ground image, its corresponding 3D scene point can be determined if the depth $d$ is known. Since acquiring an accurate depth map for the ground view is impractical, we instead hypothesize a set of $N$ reference height values $\{h_i\}_{i=1}^N$ (relative to the ground plane) for each pixel, capturing potential depth variations across different regions. For each reference height, GCA estimates an offset $\Delta h_i$ based on the current feature state and calculates an attention weight $\mathbf{A}_i$ for each height hypothesis to evaluate its reliability. The GCA mechanism then aggregates features from the satellite image as follows:

$$GCA(Q, V) = \sum_{i=1}^{N} \mathbf{A}_i \left( V \otimes P(u_g, v_g, h_i + \Delta h_i) \right), \qquad (5)$$

where $Q$ and $V$ represent the ground and satellite features, respectively. Here, $\mathbf{A}_i$ and $\Delta d_i$ are computed from the ground features $Q$, and the $\mathbf{A}_i$ is the result of applying softmax to the $N$ reference heights. $P(u_g, v_g, h_i + \Delta h_i)$ maps the ground pixel $(u_g, v_g)$ at height $h_i + \Delta h_i$ to the corresponding satellite pixel coordinates based on relative pose. The symbol $\otimes$ denotes a sampling operation that extracts features from $V$ according to the projected satellite coordinates.

Unlike previous approaches that rely strictly on predefined 3D priors (Li et al. (2024a)), our method allows for iterative refinement of satellite-to-ground correspondences. This iterative adjustment enables the model to gradually correct initial misalignments and improve spatial consistency over time. By focusing on relevant regions through projected coordinates, GCA also reduces computational load compared to traditional global attention mechanisms, effectively enhancing geometric alignment between satellite images and generating ground-level views.

## 4.2 ITERATIVE POSE ALIGNMENT DURING INFERENCE

The inherent randomness of the diffusion process and implicit reasoning can often result in generated images that deviate from the specified pose, causing positional shifts, as shown in Experiment 5.3. To address this issue, we propose an iterative position correction mechanism that adjusts each generated result to better match the specified pose throughout inference. Due to the complex scene depth, pose discrepancies can lead to substantial misalignments, making pixel-wise color or flow corrections difficult to learn. Instead, we introduce an Iterative Homography Adjustment mechanism that applies a Homography transformation matrix $H$ to correct the latent representation $z_t$ at each inference step, i.e., $z_t \otimes \text{grid}(H)$, where $\text{grid}(H)$ represents coordinates generated by $H$. The matrix $H$ enables transformations such as scaling, translation, and rotation, corresponding to the spatial alignment of the camera. Since achieving precise correction with a single $H$ is challenging, we perform iterative pose adjustments in parallel with DDIM denoising steps, gradually guiding the output toward the target pose.

The process for adjusting $H$ is illustrated in the bottom left of Fig. 2. We first compute an initial latent $z_{0,t}$ from $z_t$ using the denoising Unet's output at timestep $t$. We initialize $H$ as an identity

---

**Algorithm 1** Iterative Homography Adjustment for Pose Refinement.

---

**Input:** diffusion steps $t$, the noisy image $z_t$, satellite image conditions $\mathbf{I}_s$ and the rotation $\mathbf{R}$ and translation $\mathbf{T}$ relative to the satellite image.
**for all** $t$ from $T$ to 1 **do**
    $z_t \leftarrow z_t \otimes grid(H)$    // $H$ is a diagonal matrix of ones
    $z_{t,0}, z_{t-1} \leftarrow DDIM(z_t, t, \mathbf{R}, \mathbf{T}, \mathbf{I}_s)$    // eliminate noise to obtain $z_{t,0}, z_{t-1}$
    $H \leftarrow H - \nabla_H \mathcal{L}_{pose}(z_{t,0}, \mathbf{R}, \mathbf{T}, \mathbf{I}_s))$    // adjust the Homography matrix
    $z_{t-1} \leftarrow z_{t-1} \otimes grid(H)$
**end for**

---

matrix, applying it to $z_{0,t}$, which we then project onto an overhead view using a ground plane assumption, as in Shi et al. (2025). To ensure robust pose alignment, as illustrated in Fig. 3, we sample multiple candidate ground camera poses in addition to the specified pose, cropping and rotating the corresponding regions from the satellite image feature maps, which are extracted from the conditioning branch of the denoising Unet. We introduce a Feature Refinement module in each branch to map the ground and satellite features to a shared representation space. These modules share the same architecture but have separate weights. Next, we compute an alignment score, $\{S_k\}_{k=1}^K$, using cosine similarity between the ground and satellite feature maps at $K$ candidate poses. To quantify alignment, we use the InfoNCE loss (Oord et al. (2018)):

$$\mathcal{L}_{\text{pose}} = -\log \frac{e^{S_{k^*}/\tau}}{\sum_k e^{S_k/\tau}}, \tag{6}$$

where $\tau$ is a temperature hyperparameter. The InfoNCE loss reaches its minimum when the ground and satellite feature maps achieve the highest similarity at the ground truth (GT) relative pose $k^*$. Using this loss, we compute the gradient with respect to $H$ and update it to improve alignment.

Our Homography Adjustment module applies transformations directly to the latent vector $z_{0,t}$ rather than mapping it back to image space, avoiding the need for a heavy ground-to-satellite localization network which operates on the original image resolution. Instead, we employ a lightweight Feature Refinement module on $z_{0,t}$ and the satellite features from the denoising Unet's conditioning branch, significantly reducing the computational cost.

Training the Homography Adjustment network follows the initial training of the denoising Unet within the diffusion framework. After obtaining $z_{t,0}$ from the Unet, we then train the Homography Adjustment network using the InfoNCE loss defined in Eq. 6. During the inference stage, both the denoising Unet and the Homography Adjustment network are trained. We then use it to refine $H$ in the stochastic denoising sampling process in DDIM to improve pose alignment iteratively.

### 4.3 TEXT-GUIDED ZERO-SHOT ENVIRONMENTAL CONTROL

Many autonomous driving datasets such as CVUSA (Zhai et al. (2017)) lack detailed textual annotations. As shown in Experiment 5.4, we discovered that when using descriptions generated by a large multimodal model along with detailed satellite images as condition, the generative model tends to prioritize accurately representing the satellite image conditions, significantly diminishing the influence of textual prompts during the training process. Therefore, we separated the text prompt. Inspired by Dhariwal & Nichol (2021), we utilized text prompts to guide the direction of reasoning during the inference stage. We replace $\epsilon_t$ with $\hat{\epsilon}_t$:

$$\hat{\epsilon}_t(z_t, t, S) = \epsilon_t(z_t, t, S) + \nabla_{z_t} \mathcal{L}_{text}(z_t, c_{text}) \tag{7}$$

The gradient term $\nabla_{z_t} \mathcal{L}_{text}(z_t, c_{text})$ guides the diffusion process towards the desired direction, thereby ensuring that the generated results are consistent with the textual conditions.

To robustly obtain the loss between the generated results and the text prompt, we randomly partition $z_{t-1}$ into multiple patches and calculate the similarity with the text for each patch. We then use CLIP (Radford et al. (2021)) to extract features from the processed patches and the text, and calculate the cosine similarity between these features to obtain the final loss:

$$\mathcal{L}_{text}(z_t, c_{text}) = 1 - \frac{1}{N} \sum_{l=1}^N sim(CLIP(z_{t,0}^l), CLIP(c_{text})) \tag{8}$$

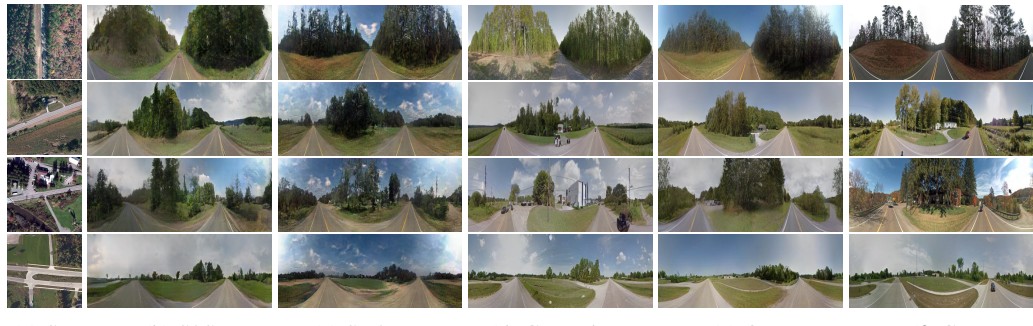

|       (a) Sat       |       (b) S2S       |       (c) Sat2Den       |       (d) ControlNet       |       (e) Ours       |       (f) GT       |

Figure 4: Qualitative comparison with previous work on the CVUSA dataset, our framework is able to maintain better geometric relationships.

Table 1: Quantitative comparison with existing algorithms on CVUSA dataset. The best results are highlighted in orange and the second-best in blue.

| Method | Structural similarity | | Perceptual similarity | | | Semantic Similarity | | Pixel Similarity | | | ↓Depth |
|---|---|---|---|---|---|---|---|---|---|---|---|
| | ↑SSIM | ↓Self_sim | ↓$P_{squeeze}$ | ↓$P_{alex}$ | ↓FID | ↓DINO | ↓SegAny | ↓RMSE | ↑PSNR | ↑SD | |
| Pix2Pix | 0.32 | 0.21 | 0.35 | 0.51 | 44.51 | 5.24 | 0.38 | 55.84 | 13.20 | 12.08 | 21.85 |
| XFork | 0.29 | - | - | - | - | - | - | - | - | - | - |
| S2S | 0.35 | 0.19 | 0.32 | 0.48 | 29.49 | 5.11 | 0.39 | 54.57 | 13.40 | 12.30 | 21.05 |
| Sat2Density | 0.33 | 0.19 | 0.31 | 0.46 | 47.85 | 4.95 | 0.38 | 54.23 | 13.46 | 12.27 | 19.83 |
| ControlNet | 0.32 | 0.20 | 0.33 | 0.49 | 22.55 | 5.21 | 0.38 | 58.25 | 12.83 | 12.08 | 21.78 |
| CrossDiff | 0.37 | - | - | - | 23.67 | - | - | - | 12.00 | - | - |
| Ours | **0.38** | **0.18** | **0.30** | **0.45** | **21.30** | **4.81** | **0.36** | **52.92** | **13.67** | **12.33** | **19.58** |

where $z_{t,0}$ is the result of removing noise from $z_t$, $z_{t,0}^l$ is the $l$-th patch of $z_{t,0}$, $c_{text}$ is the textual condition, and $sim$ represents cosine similarity. Similar to the Iterative Pose Alignment illustrated in Sec. 4.2, this environmental control strategy is also applied during the inference stage, iteratively modifying the noisy latent during the denoising process.

## 4.4 EVALUATION METRICS

There is a limited overlap between satellite images and ground truth images, further complicated by varying weather conditions and seasons, which makes it challenging to synthesize target view images that perfectly match the ground truth image provided in the dataset. As a result, pixel-level metrics like RMSE and PSNR are inadequate for this task (Zhang et al. (2018); Shi et al. (2022a)). Instead of focusing on color discrepancies between the generated street view images and the actual ground truth images, our emphasis should be on whether they represent the same locations. Therefore, we use structural similarity, perceptual similarity, semantic similarity, and depth similarity for performance evaluation. Furthermore, since satellite images do not capture sky information, we advocate for cropping the sky portion during evaluation.

The evaluation of Structural Similarity is derived from SSIM and self-similarity structures based on DINO features (Caron et al. (2021); Shechtman & Irani (2007); Tumanyan et al. (2022)). Perceptual Similarity is evaluated based on the FID metric and compares the similarity of features extracted from AlexNet (Krizhevsky et al. (2017)) and SqueezeNet (Iandola (2016)). Semantic Similarity is proposed for evaluating high-level semantic features. We employ widely acknowledged DINO (Caron et al. (2021)) and Segment Anything (Kirillov et al. (2023)) for feature extraction to compare the semantic consistency of images. Additionally, given that satellite images contain depth information of ground scenes, evaluating the depth of generated ground images is crucial. We use DepthAnything (Yang et al. (2024)) to assess the depth differences between real ground truth images and generated images. For a fair comparison, we also report commonly used metrics such as RMSE, PSNR, and SD to evaluate Pixel Similarity. These metrics are more forgiving for assessing blurred images, as clear images tend to exhibit increased pixel differences due to the introduction of details such as seasons, weather, and shadows. This discrepancy conflicts with the intended outcome of our generated images, so we strongly recommend utilizing alternative metrics over pixel similarity evaluation metrics. Finally, for text similarity, we compute the similarity of Clip Radford et al. (2021)

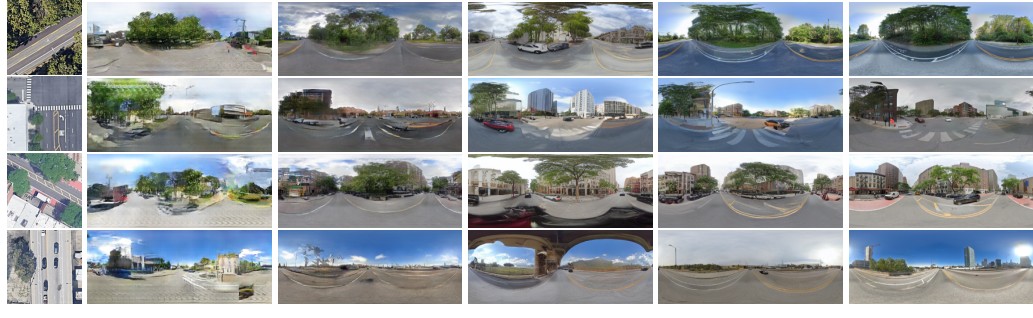

| (a) Sat | (b) S2S | (c) Sat2Den | (d) ControlNet | (e) Ours | (f) GT |

Figure 5: Qualitative comparison with previous work on the VIGOR dataset, our model can effectively capture road surface information from satellite images, resulting in clearer and more distinct road lines.

Table 2: Quantitative comparison with existing algorithms on VIGOR dataset. The best results are highlighted in orange and the second-best in blue.

| Method | Structural similarity | | Perceptual similarity | | | Semantic Similarity | | Pixel Similarity | | | ↓Depth |
|---|---|---|---|---|---|---|---|---|---|---|---|
| | ↑SSIM | ↓Self_sim | ↓$P_{squeeze}$ | ↓$P_{alex}$ | ↓FID | ↓DINO | ↓SegAny | ↓RMSE | ↑PSNR | ↑SD | |
| Pix2Pix | 0.37 | 0.17 | 0.33 | 0.45 | 67.96 | 4.72 | 0.38 | 55.00 | 13.33 | 12.93 | 8.65 |
| S2S | 0.33 | 0.18 | 0.37 | 0.49 | 121.10 | 5.03 | 0.40 | 62.94 | 12.16 | 12.31 | 10.87 |
| Sat2Density | 0.40 | 0.16 | 0.28 | 0.39 | 54.49 | 4.41 | 0.36 | 50.23 | 14.14 | 12.90 | 8.05 |
| ControlNet | 0.34 | 0.17 | 0.34 | 0.46 | 23.68 | 4.95 | 0.39 | 63.98 | 12.02 | 12.59 | 10.01 |
| Ours | 0.42 | 0.16 | 0.27 | 0.38 | 28.01 | 4.34 | 0.35 | 52.16 | 13.80 | 13.07 | 7.10 |

features for evaluation and utilize Blip Li et al. (2022) to describe the images, assessing the recall rate of answers that align with the textual descriptions.

## 5 EXPERIMENTS

### 5.1 IMPLEMENTATION DETAILS

**Experimental setup.** We take $256 \times 256$ satellite images as input to predict $128 \times 512$ ground images, following the same setup as in Shi et al. (2022a) for fair comparison. Our model is finetuned based on the Stable Diffusion 1.5 model (Rombach et al. (2022a)), with the Cross-Attention of diffusion replaced by Geometric Cross-Attention, and satellite image conditions processed through a simple VIT network for feature extraction. During inference, we employ DDIM sampling with 50 sampling steps, applying the Homography Adjustment scheme in the first 40 sampling steps and Zero-Shot Environmental Control throughout the entire sampling process. In Geometric Cross-Attention, we utilize 8 sampling heights of [-3, -2, -1, 1, 2, 3, 4, 5]. This constitutes an empirical setup. Training process on three GPUs with batch size of 32 for 200 epochs.

**Datasets.** We adopt three cross-view datasets: CVUSA (Zhai et al. (2017)), KITTI (Geiger et al. (2013); Shi & Li (2022)), and VIGOR (Zhu et al. (2021); Lentsch et al. (2022)). These datasets comprise pairs of cross-view data, combining ground-level images with their corresponding satellites. The ground-level images in CVUSA and VIGOR are panoramic, while the ground-level images in KITTI have a limited horizontal field of view (HFoV). CVUSA comprises 35,532 pairs of satellite and street view images for training and 8,884 pairs for testing. Following the setup of the cross-view localization task (Shi & Li (2022); Xia et al. (2023)), KITTI includes 19,655 pairs in the training data and 3,773 pairs in the testing data. VIGOR gathers data from New York, Seattle, San Francisco, and Chicago, dividing the data from each city into 52,609 pairs for the training set and 52,605 pairs for the test set.

### 5.2 COMPARISON WITH EXISTING METHODS

In this section, we compare our approach with previous ground map generation methods. Pix2Pix (Isola et al. (2017)) and XFork (Regmi & Borji (2018)) are GAN-based methods that extract implicit features from satellite images to generate ground images. S2S (Shi et al. (2022a)) and Sat2Den (Qian et al. (2023)) introduce explicit geometric information into the network and project satellite images to ground view based on height priors. CrossDiff (Li et al. (2024a)) is a diffusion

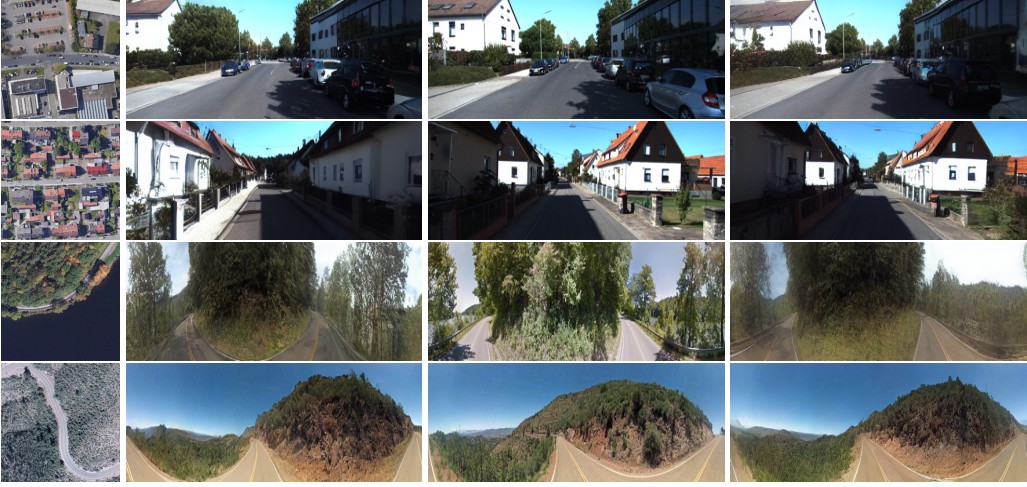

(a) Sat Map         (b) LDM           (c) GT           (d) LDM W.IHA

Figure 6: Qualitative ablation experiments of IHA on KITTI and CVUSA datasets. IHA can correct pose offsets in the inference process effectively.

Table 3: Ablation study on the KITTI dataset. We compared the positional accuracy of generated images using a cross-view localization model.

| | Distance | | Angle | | Lateral | | Longitudinal | | Azimuth | |
|---|---|---|---|---|---|---|---|---|---|---|
| | ↓average | ↓median | ↓average | ↓median | ↑d=1 | ↑d=3 | ↑d=1 | ↑d=3 | ↑θ=1 | ↑θ=3 |
| LDM | 10.74 | 6.92 | 17.60 | 6.64 | 43.97 | 67.74 | 15.35 | 36.79 | 7.87 | 24.04 |
| LDM W. IHA | 10.67 | 6.85 | 17.46 | 6.73 | 44.08 | 67.77 | 15.77 | 36.95 | 8.08 | 23.78 |
| LDM W.GCA W.IHA | **10.51** | **6.66** | **17.14** | **6.64** | **45.40** | **68.67** | **16.78** | **38.22** | **8.31** | **24.22** |

method that relies on high prior knowledge. Furthermore, noticing the robust capabilities of Control-Net (Zhang et al. (2023)), we also compared it with our method. The implementation of ControlNet mirrors the successful case (Sastry et al. (2024)). ControlNet receives textual conditions and ground map contour conditions. The textual conditions are generated using LLAVA (Liu et al. (2024)) and randomly masked out with a probability of 0.5 during training. The ground map contour conditions are derived from satellite maps projected based on prior ground height assumptions.

Quantitative results, as shown in Table 1 and Table 2, demonstrate that the quality of ground image generation produced by our method significantly surpasses that of other approaches, particularly in maintaining geometric consistency. This performance stems from the Cross-View Conditioning Mechanism we employ and the Iterative Pose Alignment conducted during inference. In the visual representation, as depicted in Fig. 4, our method adheres to the geometric cues from satellite images, generating ground images that align with the scenes and outperform previous algorithms in representing pathways. While ControlNet outperforms us in the FID metric on the VIGOR dataset, we note that its other quantitative metrics are notably lower, indicating that ControlNet accurately captures dataset distributions but struggles to faithfully translate satellite image hints into improved ground image generation, a point supported by Fig. 5. On the VIGOR dataset, our method excels in ground feature identification and architectural representation compared to other algorithms.

## 5.3 THE EFFECTIVENESS OF POSE ALIGNMENT

Compared to other datasets, KITTI's data is collected along the same route and divided into training and test sets. Due to the high similarity in satellite image conditions between adjacent points along the path, the implicit condition diffusion model fails to distinguish effectively, leading to the generation of ground images with positional offsets quite easily. To demonstrate the effectiveness of the proposed pose alignment approaches in this paper, we employ the original implicit diffusion model as our baseline and gradually add the proposed modules, i.e., the IHA and the GCA. To better evaluate the results of Pose Alignment, we employ CCVPE (Xia et al. (2023)), a powerful cross-view pose estimation method, to evaluate the consistency of the relative pose between generated images and the satellite images with respect to the conditioning pose, in addition to image-level similarities.

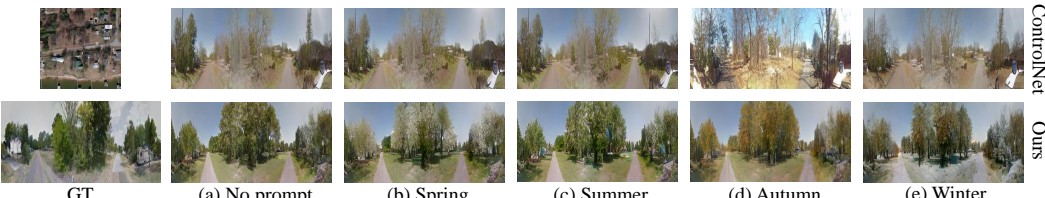

|  |  |  |  |  | ControlNet |
|  |  |  |  |  | Ours |
| GT | (a) No prompt | (b) Spring | (c) Summer | (d) Autumn | (e) Winter |

Figure 7: Qualitative comparison with ControlNet on CVUSA. ControlNet trained under weak textual conditions fails to effectively adjust image information based on text prompts. Our Zero-shot Environmental Control can adjust image information based on text prompts while preserving the spatial structure of the image.

Table 4: The similarity between generated images and environmental text prompt.

|  | Clip Score | | | | Blip Score | | | |
|  | spring | summer | autumn | winter | spring | summer | autumn | winter |
| --- | --- | --- | --- | --- | --- | --- | --- | --- |
| LDM | 0.21 | 0.21 | 0.20 | 0.18 | 5.14 | 60.87 | 7.44 | 1.78 |
| ControlNet | 0.21 | 0.21 | 0.20 | 0.20 | 12.89 | 44.63 | 5.50 | 13.06 |
| Ours | **0.26** | **0.24** | **0.24** | **0.23** | **69.74** | **82.05** | **78.62** | **47.67** |

The quantitative comparison is shown in Table 3. It can be seen that both IHA and GCA improve the generated image quality in terms of image-level similarity and pose consistency.

Fig. 6 provides examples of Pose Alignment on KITTI and CVUSA datasets. Both the images before and after correction are generated with the same noise, the only difference being whether Pose Alignment is enabled. We can observe the significant role played by Pose Alignment in successfully correcting pose deformities in the generated results. Furthermore, the method is equally applicable to both FoV images and panoramic images, demonstrating a high level of universality.

## 5.4 THE EFFECTIVENESS OF TEXT-GUIDED ENVIRONMENTAL CONTROL

To evaluate the effectiveness of the Text-guided Environmental Control mechanism, we conduct comparative experiments with LDM and ControlNet on the CVUSA dataset, generating images representing the four seasons: spring, summer, autumn, and winter. Notably, our environmental control mechanism is applied during inference in a zero-shot manner, while for LDM and ControlNet, the environmental control is incorporated as an additional text condition embedded within the denoising network. To train ControlNet's text condition branch, we used LLAVA to generate text annotations for the ground images.

As shown in Table 4, the quantitative results indicate that our method produces outputs that align more effectively with the provided textual prompts. One challenge we encountered was the inconsistency in the quality of LLAVA-generated text descriptions, as they were produced without manual annotations. During ControlNet's training, the model tended to prioritize well-expressed satellite image prompts, diminishing the impact of textual prompts over time. This behavior led to ControlNet's reduced sensitivity to diverse text prompts, as illustrated in Fig. 7. In contrast, our Zero-shot Environmental Control (ZoEC) mechanism exhibited greater robustness in handling unsupervised scenarios, maintaining stronger adherence to the given text conditions.

## 6 CONCLUSION

In this paper, we have presented a novel approach for generating ground-level images from satellite imagery, addressing the dual challenges of geometric alignment and environmental diversity. Our method introduced two key innovations: the Iterative Homography Adjustment (IHA) mechanism, which ensures accurate pose alignment between the satellite and generated ground-level views, and the Text-guided Zero-shot Environmental Control (ZoEC), which allows flexible control over lighting, weather, and seasonal variations without requiring additional training data. By incorporating geometric cross-attention in the diffusion process, we further improved the consistency between satellite and street-level views. Extensive experiments demonstrated that our method outperforms existing approaches in both geometric accuracy and environmental adaptability.

ACKNOWLEDGMENTS

The authors are grateful for the valuable comments and suggestions by the reviewers and AC. This work was supported by NSFC (62406194, 62302220), Shanghai Frontiers Science Center of Human-Centered Artificial Intelligence (ShangHAI), MoE Key Laboratory of Intelligent Perception and Human-Machine Collaboration (KLIP-HuMaCo). A part of the experiments of this work were supported by the core facility Platform of Computer Science and Communication, SIST, ShanghaiTech University.

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

# A APPENDIX

## A.1 THE ROBUSTNESS OF IHA IN THE FACE OF NOISE.

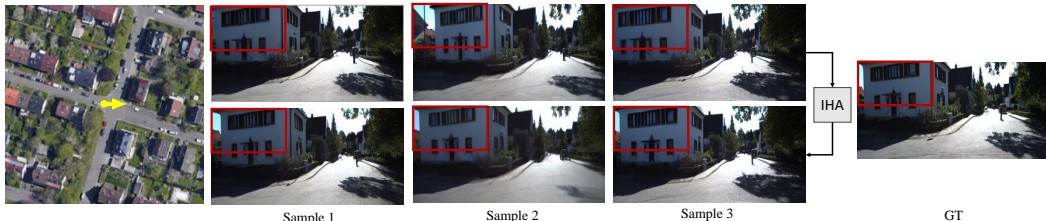

Figure 8: In the first row, we demonstrate that ground images generated from satellite maps exhibit varying offsets under different initial noise conditions. The second row illustrates the pose correction results of the first row using the Iterative Homography Adjustment (IHA).

As shown in Fig. 8, when using different initial noises, the reverse diffusion process often produces unpredictable outcomes. In the first row, diverse noises result in different view perspectives. However, the IHA module adeptly corrects these deviations and gets consistent results. This case shows the remarkable robustness of IHA for identifying and rectifying the accurate pose of generated ground images.

## A.2 DATA AUGMENTATION FOR CROSS-VIEW LOCALIZATION

Table 5: Using generated street data for data augmentation. We report the localization accuracy of Shi et al. (2025) on the KITTI dataset.

| Test1 | | | | | | | | | | |
|---|---|---|---|---|---|---|---|---|---|---|
| | | Distance | | Angle | | Lateral | | Longitudinal | | Azimuth |
| | | ↓average ↓median | | ↓average ↓median | | ↑d=1 ↑d=3 | | ↑d=1 ↑d=3 | | ↑θ=1 ↑θ=3 |
| Wo.augmentation | | 11.11 | 7.646 | 0.1811 | 0.1492 | 56.98 87.23 | | 10.68 31.57 | | **99.66 100.0** |
| W.augmentation | Pix2Pix | 11.72 | 9.020 | **0.1810** | **0.1491** | 54.31 86.03 | | 9.833 27.54 | | **99.66 100.0** |
| | ControlNet | 11.15 | 7.688 | **0.1810** | **0.1491** | 54.17 87.36 | | 11.61 31.27 | | **99.66 100.0** |
| | Ours | **10.88** | **7.167** | **0.1810** | **0.1491** | **57.57 87.70** | | **11.95 32.65** | | **99.66 100.0** |

| Test2 | | | | | | | | | | |
|---|---|---|---|---|---|---|---|---|---|---|
| | | Distance | | Angle | | Lateral | | Longitudinal | | Azimuth |
| | | ↓average ↓median | | ↓average ↓median | | ↑d=1 ↑d=3 | | ↑d=1 ↑d=3 | | ↑θ=1 ↑θ=3 |
| Wo.augmentation | | 14.07 | 10.61 | 0.1570 | **0.1305** | 60.32 84.90 | | 8.632 25.14 | | **100.0 100.0** |
| W.augmentation | Pix2Pix | 14.44 | 11.29 | **0.1569** | **0.1305** | 57.44 80.63 | | 8.101 23.89 | | **100.0 100.0** |
| | ControlNet | 14.31 | 10.91 | **0.1569** | **0.1305** | 55.99 82.39 | | 8.260 25.05 | | **100.0 100.0** |
| | Ours | **13.94** | **9.885** | **0.1569** | **0.1305** | **62.87 85.51** | | **8.923 26.92** | | **100.0 100.0** |

The cross-view positioning algorithm has achieved a high level of accuracy after a long period of development (Shi et al. (2019; 2020a;b; 2022b;c; 2023); Song et al. (2024); Xia et al. (2022; 2025)). To verify that the proposed generative model can assist in vehicle localization in autonomous driving scenarios, we utilize generated street data for data augmentation in training a localization model. The experiments are conducted on the KITTI dataset (Geiger et al. (2013); Shi & Li (2022)). Geiger et al. (2013) provides ground images captured by vehicles, while Shi & Li (2022) collects corresponding satellite images for each ground image. The dataset is divided into Training, Test1, and Test2 subsets. The images in Test1 come from the same area as the images in the training set, while the images in Test2 come from different areas. In our data augmentation experiment, we use a ground map resolution of 128x512 and a satellite map resolution of 512x512. For each satellite image in the training set of the KITTI dataset, we utilize the proposed framework to generate street scene images under various season conditions. During the training of the cross-view localization algorithm(Shi et al. (2025)), both generated and real data are employed with a 50% probability each to augment the KITTI dataset. In Table 5, we present the localization accuracy with and without the

data augmentation. Both of them are trained for 15 epochs. In the W.augmentation experiments, we compare the data generated by Pix2Pix, ControlNet, and our own method. The data augmentation with our generated data significantly improves cross-view localization performance, demonstrating the usefulness of our approach to autonomous driving tasks. Compared to our generation strategy method, the data generated by GAN-based Pix2Pix is very blurry. Adding the generated data from Pix2Pix actually deteriorates the model's perception of details. The data generated by ControlNet, although clear, lacks strong positional constraints. The pose offset leads to incorrect matches in the localization model during training. Compared to other methods, our approach can ensure geometrically consistent generation and text-guided diverse environment creation, thereby achieving the purpose of data augmentation and yielding outstanding results.

### A.3 MORE ANALYSIS OF THE IHA.

#### A.3.1 THE HOMOGRAPHY TRANSFORMATION IN IHA.

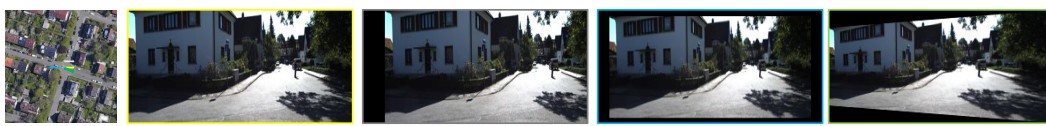

Figure 9: The relationship between Homography Adjustment and the camera position in satellite imagery. The corresponding satellite position and ground images are labeled with arrows and borders using the same colors.

The Homography transformation is defined by a 3x3 matrix and can map a plane in an image to another plane. It allows for image manipulations including rotation, translation, scaling, shearing, and perspective transformation. As depicted in Fig. 9, adjusting the ground map through Homography enables correspondence with different perspectives in a satellite image. Translating the ground image corresponds to perpendicular movements to the satellite image, scaling represents horizontal (front-back) movements within the satellite image, and the projection transformation signifies changes in yaw angle.

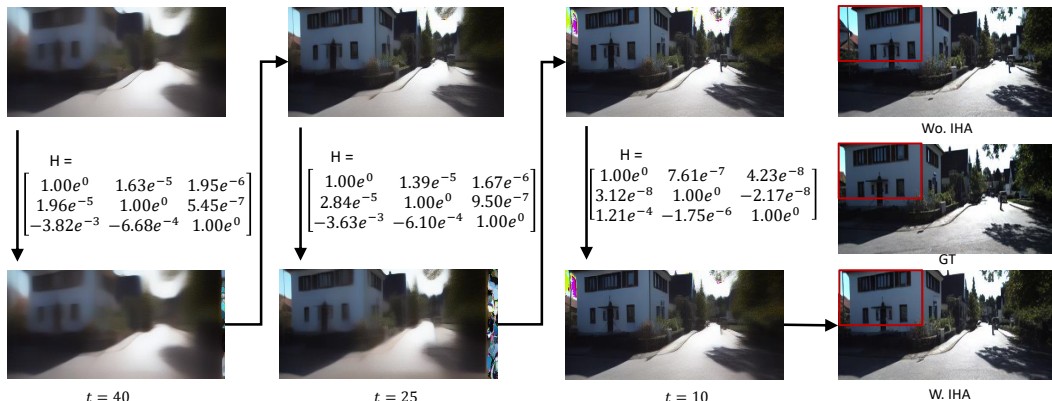

Figure 10: The Homography Adjustment operation in the DDIM process. The visual results of each step stem from $\mathcal{D}(z_{t,0})$
.

Due to variations in initial noise and uncertainties in the implicit learning of features, diffusion does not always yield favorable outcomes. As depicted in Fig. 6, many ground images may become distorted, indicating a displacement in the corresponding satellite image coordinates. Motivated by this, we employ satellite image-guided supervision in the DDIM generation process. Within the DDIM process, we utilize satellite image conditions, the result $z_{t-1}$ at step $t$, and the expected location $R$, $T$ of the ground image in the satellite image. We supervise the positioning algorithm illustrated in Fig. 3 to adjust the location of the generated $z_{t-1}$ based on the Homography matrix for the ground image. Given the substantial difficulty in achieving correct results through a single Homography Adjustment, we adopt an iterative approach, progressively executing Algorithm 1 to guide the ground image pixels toward the correct coordinates. In Fig. 10, we illustrate the intermediate results and step-by-step changes in the denoising process using IHA.

A.3.2 THE COMPUTATIONAL WORKLOAD IN IHA

Table 6: The analysis of computational efficiency of IHA.

|  | Memory | Time Cost |
|---|---|---|
| Wo.IHA | 20126MB | 5.406s |
| W.IHA | 21022MB | 5.513s |

The IHA operates in the latent space using low-resolution feature maps. Since the IHA calculation relies on the pixel coordinates, it does not impose a significant computational increase. We conduct the efficiency tests with a batch size of 1, and the results are presented in Table 6. When comparing memory usage, the baseline model without IHA (Wo.IHA) requires 20,126MB, while the inclusion of IHA (W.IHA) increases memory usage slightly to 21,022MB (896MB increase) due to the addition of a lightweight localization network. In terms of time cost, the baseline model (Wo.IHA) requires 5.406 seconds per image, while the model with IHA (W.IHA) increases this slightly to 5.513 seconds per image—an additional cost of only 0.107 seconds per image. We believe that IHA strikes a favorable balance between performance and computational efficiency, offering promising practical value.

A.4 DERIVATION OF FORMULAE.

Based on the score-based formulation of a diffusion modelSong et al. (2020), and the introduction of conditional guidance $g_{pose}, g_{text}$, our objective is to learn

$$\hat{\epsilon}_t = -\sqrt{1 - \bar{\alpha}_t} \nabla_{z_t} \log p(z_t | g_{pose}, g_{text}) \tag{9}$$

Using the Bayes' formula, we obtain:

$$p(z_t | g_{pose}, g_{text}) = \frac{p(g_{pose}, g_{text} | z_t) p(z_t)}{p(g_{pose}, g_{text})} \tag{10}$$

In the formula, $p(g_{pose}, g_{text})$ can be considered a constant, and we denote it as $C$. Utilizing the independence of environmental and positional conditions, we express the formula as:

$$p(z_{t-1} | g_{pose}, g_{text}) = C p(g_{pose}, g_{text} | z_t) p(z_t)$$
$$= C p(g_{pose} | z_t) p(g_{text} | z_t) p(z_t) \tag{11}$$

We substitute Eq.11 into Eq.9 to obtain:

$$\hat{\epsilon}_t = -\sqrt{1 - \bar{\alpha}_t} \nabla_{z_t} \log p(z_t |, g_{pose}, g_{text})$$
$$= -\sqrt{1 - \bar{\alpha}_t} (\nabla_{z_t} \log p(z_t) + \nabla_{z_t} \log p(g_{pose} | z_t) + \nabla_{z_t} \log p(g_{text} | z_t)) \tag{12}$$

From the score-based formula $\epsilon_t = -\sqrt{1 - \bar{\alpha}_t} \nabla_{z_t} \log p(z_t)$, we can derive:

$$\hat{\epsilon}_t = \epsilon_t - \sqrt{1 - \bar{\alpha}_t} \nabla_{z_t} \log p(g_{pose} | z_t) - \sqrt{1 - \bar{\alpha}_t} \nabla_{z_t} \log p(g_{text} | z_t) \tag{13}$$

Substituting the new $\hat{\epsilon}_t$ into Eq. 3 in place of $\epsilon_\theta(z_t, t, c)$, we obtain:

$$z_{t-1} = \mu_\theta(z_t, t, c) + \gamma \nabla_{z_t} \log p(g_{\text{text}} | z_t) + \lambda \nabla_{z_t} \log p(g_{\text{pose}} | z_t) + \sigma_t \varepsilon \tag{14}$$

The hyperparameters $\gamma$ and $\lambda$ control the influence of pose and environmental conditioning, where higher values strengthen the alignment with desired conditions.

A.5 QUALITATIVE ANALYSIS OF RURAL SCENES PERTAINING TO CVUSA.

Fig. 11 presents additional visual results using the CVUSA dataset, mostly collected in rural areas. Notably, S2S and Sat2Den struggle to capture the geometric features of rural scenes, producing blurry and indistinct images. ControlNet also shows noticeable geometric misalignments. In contrast, our method, leveraging the proposed Geometric Cross-Attention (GCA) mechanism and Iterative Homography Adjustment (IHA), demonstrates superior recovery of road structures.

For the CVUSA dataset, another characteristic is that buildings often occupy only a small region in the target ground-view images. The latent embedding of conditioning satellite images can hardly

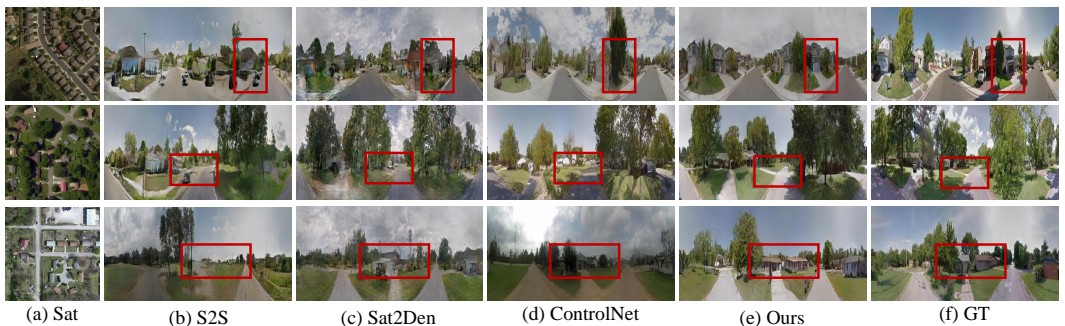

| (a) Sat | (b) S2S | (c) Sat2Den | (d) ControlNet | (e) Ours | (f) GT |

Figure 11: Qualitative visualization for generated images using different methods in complex scenes.

encode the buildings due to few occupant pixels. Owing to the geometric reasoning capabilities of GCA and IHA, our method excels at mining deep semantic features at the correct coordinates. As a result, it accurately reconstructs buildings in the appropriate regions of the ground-view images and synthesizes realistic facade appearances, as illustrated in the third example.

In summary, our method not only retains strong structural consistency but also exhibits a promising ability to generate small elements in satellite images.

## A.6 MORE ANALYSIS OF THE GCA.

### A.6.1 VISUAL ANALYSIS OF THE GCA MECHANISM.

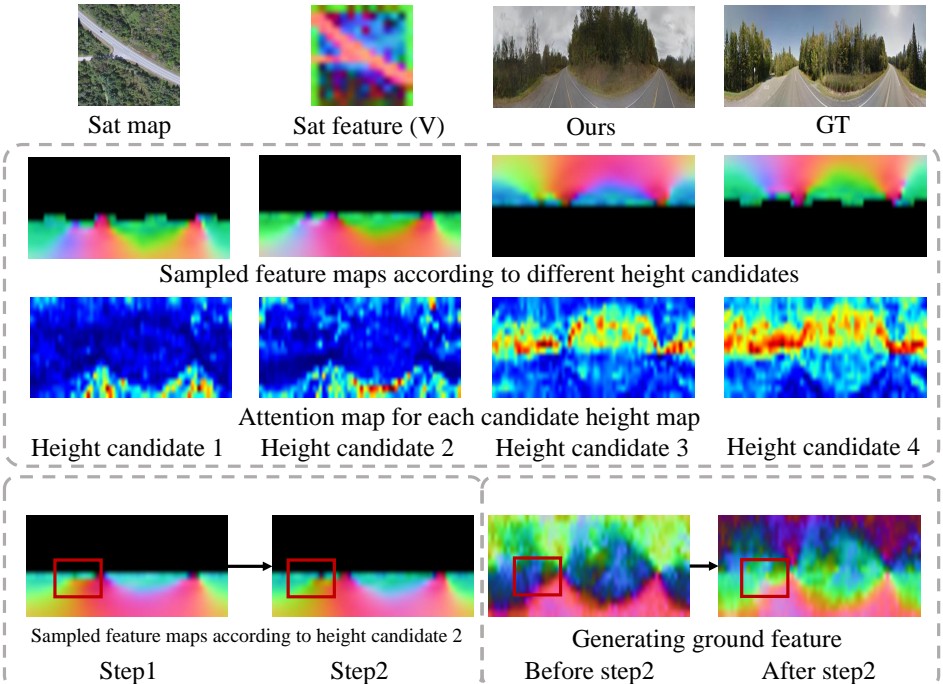

Figure 12: Visualization results of the GCA mechanism. The feature maps are visualized using PCA, while the attention maps are displayed with a jet color mapping. In the middle row, we present ground feature maps sampled from satellite features according to four candidate height map candidates alongside their corresponding attention maps. On the bottom left, we illustrate ground feature maps sampled from satellite features according to the same height plane but at different GCA steps, with feature map differences reflecting variations in height information. The bottom right shows the aggregated features obtained from the learned height candidates across different GCA steps, demonstrating the progressive refinement achieved by the GCA mechanism.

In this section, we present visualizations of intermediate features of satellite images under varying height assumptions and the corresponding attention maps $A_i$. The progressive ground features using the GCA mechanism are also revealed.

During the GCA process, the inputs include the current ground map features ($Q$), satellite map features ($V$), and the relative poses $R$ and $T$. We set eight height assumptions ($h \in \{-3, -2, -1, 1, 2, 3, 4, 5\}$ in experiments. Here, $h$ represents the relative height of the satellite image to the camera. When the scene lies below the camera, the lower part of the camera captures the scene while the camera's upper part works for scenes above the camera. This behavior is illustrated in the second row of Fig. 12.

As defined in Eq. 5, the GCA mechanism infers offsets $\Delta h_i$ and attention weights $A_i$ for each height candidate based on the current ground map features ($Q$). The attention weights $A_i$ are normalized via a softmax function across the eight height planes, determining the confidence associated with each plane.

In the middle of Fig. 12, the top row shows the ground feature maps sampled from satellite features at different candidate heights ($h_i + \Delta h_i$), while the bottom row visualizes the corresponding attention maps $\{A\}_{i=1}^N$. In the first GCA iteration (step 1), the projection plane samples features based on the initial height assumption, and the attention is evenly distributed across various regions, providing a baseline solution. By the second GCA iteration (step 2), the inferred offsets $\Delta h_i$ introduce positional shifts in the projection plane, as highlighted in the red box. These shifts enable the projection to capture more details of ground images, such as pathways. The GCA iteration also gradually refines the representations of other scene elements.

As the GCA iterations progress, the attention maps evolve to focus on salient regions. For planes where $h < 0$, the attention primarily targets the ground, while for planes where $h > 0$, it shifts towards elevated elements like trees. The attention shifting illustrates that GCA effectively enhances feature representations of the entire scene in a progressive manner.

### A.6.2 COMPARISON WITH NAIVE CROSS-ATTENTION.

The proposed GCA module offers the following advantages over a naive cross-attention mechanism.

1. Flexibility in view-related image generation. By leveraging the relative pose between satellite and ground images, the proposed GCA enables the generation of ground images at arbitrary locations and from arbitrary views on the same satellite map. In contrast, the simple cross-attention mechanism cannot handle view changes and requires additional modules to process the relative pose information (e.g., Zero-1-to-3 Liu et al. (2023)). Our method is more flexible and can handle various relative pose differences.

2. Avoid redundancy information. The proposed GCA limits attention to regions likely to correspond geometrically. Each ground-view pixel only attends to satellite image areas along its camera ray. This focused attention minimizes noise from irrelevant regions, unlike naive cross-attention, which indiscriminately considers the entire image.

3. Improved computational efficiency and reduced GPU memory usage. GCA achieves significant reductions in computational complexity by employing sparse sampling. For satellite image features of size $S * S$ and ground image features of size $H * W$, the complexity of naive cross-attention is $O(S * S * H * W)$. In contrast, our algorithm samples N planes (N=8), with the complexity of sampling being $O(N * H * W)$, computing horizontal and vertical coordinate offsets at $O(2N * H * W)$, and attention calculation at $O(N * H * W)$. Overall, our complexity is significantly reduced to $O(4 * N * H * W)$ compared to naive cross-attention. In practical experiments, for inference on one example, native cross-attention consumes 250MB of GPU memory, whereas using GCA reduces this to only 184MB.

### A.7 DISCUSSING THE REASONS FOR THE FAILURE OF THE CFG SCHEME IN CONTROLLING THE ENVIRONMENT.

In Tab 4, we employ the LDM and ControlNet models using a Classifier-Free Guidance (CFG) approach for multi-condition generation. Both LDM and ControlNet take two conditions as inputs: satellite images and image descriptions generated by LLAVA. Through training, we observed that

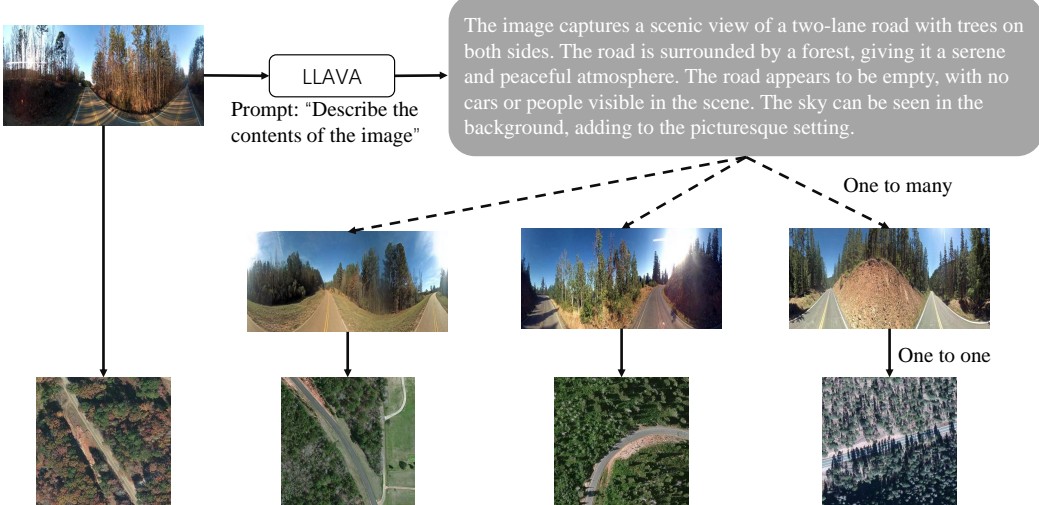

Figure 13: Using LLAVA-generated text for ground images can correspond to multiple ground pictures, while satellite images correspond one-to-one with ground images.

the weight assigned to the satellite image condition far exceeded that of the environmental description condition. Furthermore, as depicted in Fig. 13, one image description generated by LLAVA could correspond to multiple ground images, and text captions significantly lack geometric information. Each ground image corresponds specifically to a satellite image, which not only encapsulates geometric descriptions but also texture representations. Consequently, the LDM and ControlNet tend to prioritize the satellite images over the ground image descriptions, leading to the degraded control of environmental conditions.

### A.8 THE PERFORMANCE OF TEXT-GUIDED ZERO-SHOT ENVIRONMENTAL CONTROL IN A MULTILINGUAL SETTING.

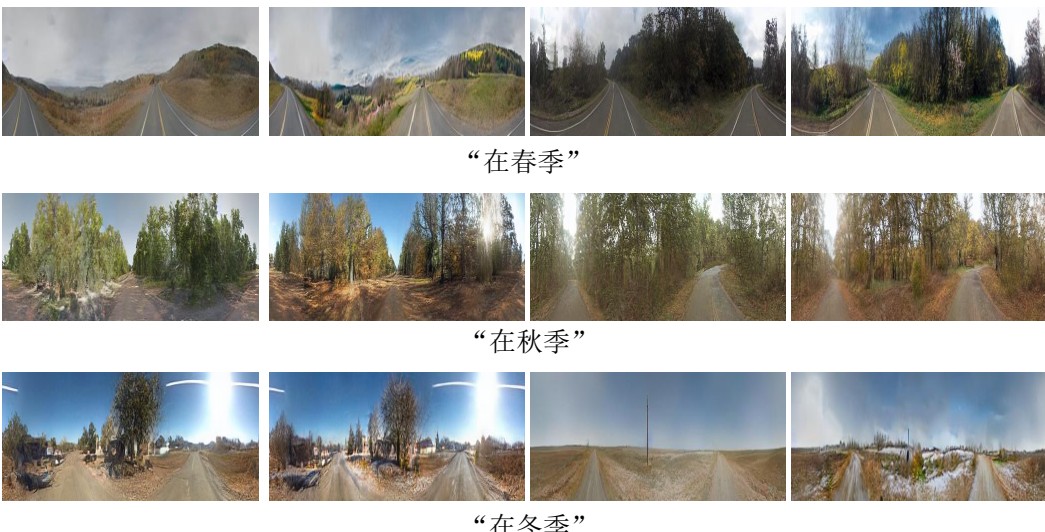

Figure 14: Controllable generation in the Chinese context. This figure displays generated images of spring, autumn, and winter from top to bottom.

In order to ensure compatibility across linguistic contexts, our framework provides a flexible interface that can integrate with various multimodal large language models. In the paper implementation, we utilize a frozen CLIP model to extract textual and visual features and compute losses. This approach eliminates additional training and enables straightforward adaptation to other multimodal

LLMs. For instance, to support Chinese text prompts, Chinese-CLIP (Yang et al. (2022)) is incorporated. By computing losses between Chinese text and image features, the image-generating process can be effectively guided to align with the input language.

## A.9 DISCUSSION AND LIMITATIONS

The primary issue revolves around the accurate restoration of building details, which is a common limitation in the task of ground image generation from satellite data. Due to the significant discrepancy between satellite and ground views, satellite images lack the texture details of building facades. Therefore, it is difficult to faithfully reconstruct the building details from the ground view. The other limitation is the lane marking generation. There are inconsistencies between the generated lanes and the real-world lanes. These differences arise largely due to misalignment of the capture time between the satellite maps and ground images. Despite these limitations, our method demonstrates significant improvements in terms of geometric alignment and environmental control in generating diverse ground-view images.

## A.10 THE RESULTS OF TEXT-GUIDED ZERO-SHOT ENVIRONMENTAL CONTROL.

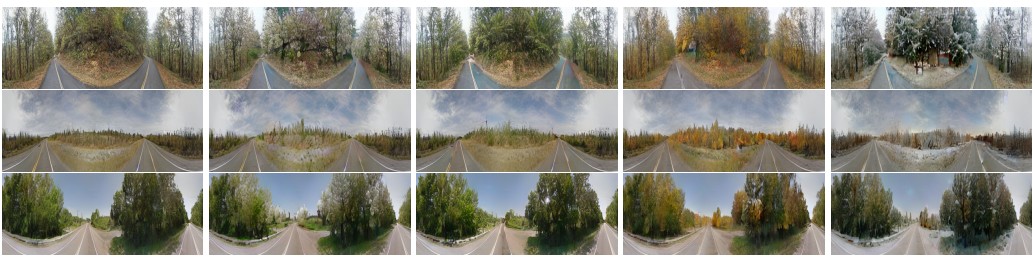

(a) No text prompt    (b) Spring    (c) Summer    (d) Autumn    (e) Winter

Figure 15: Generating under different textual conditions to obtain text-guided image results while maintaining the structure of the images

## A.11 MORE RESULTS FROM POSE ALIGNMENT.

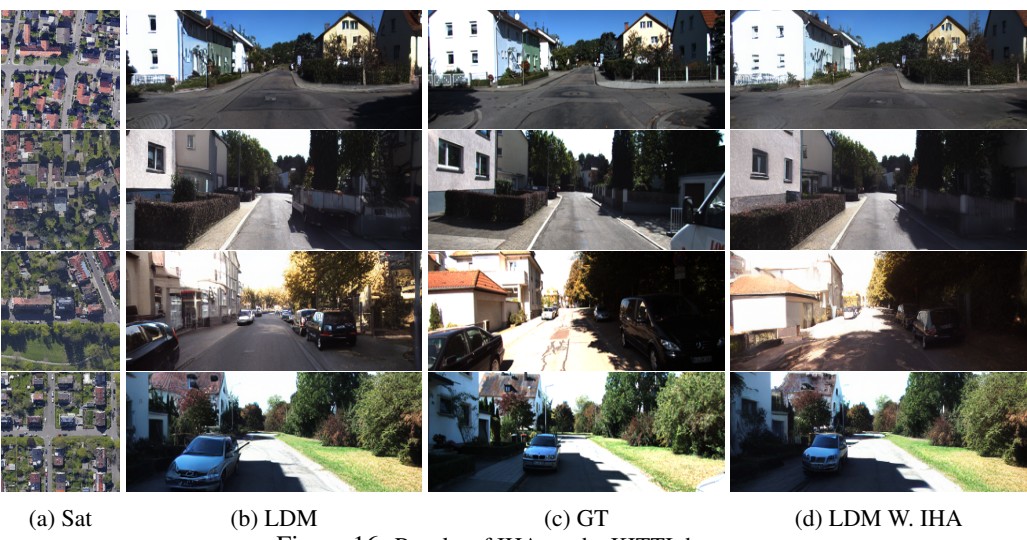

(a) Sat    (b) LDM    (c) GT    (d) LDM W. IHA

Figure 16: Results of IHA on the KITTI dataset.

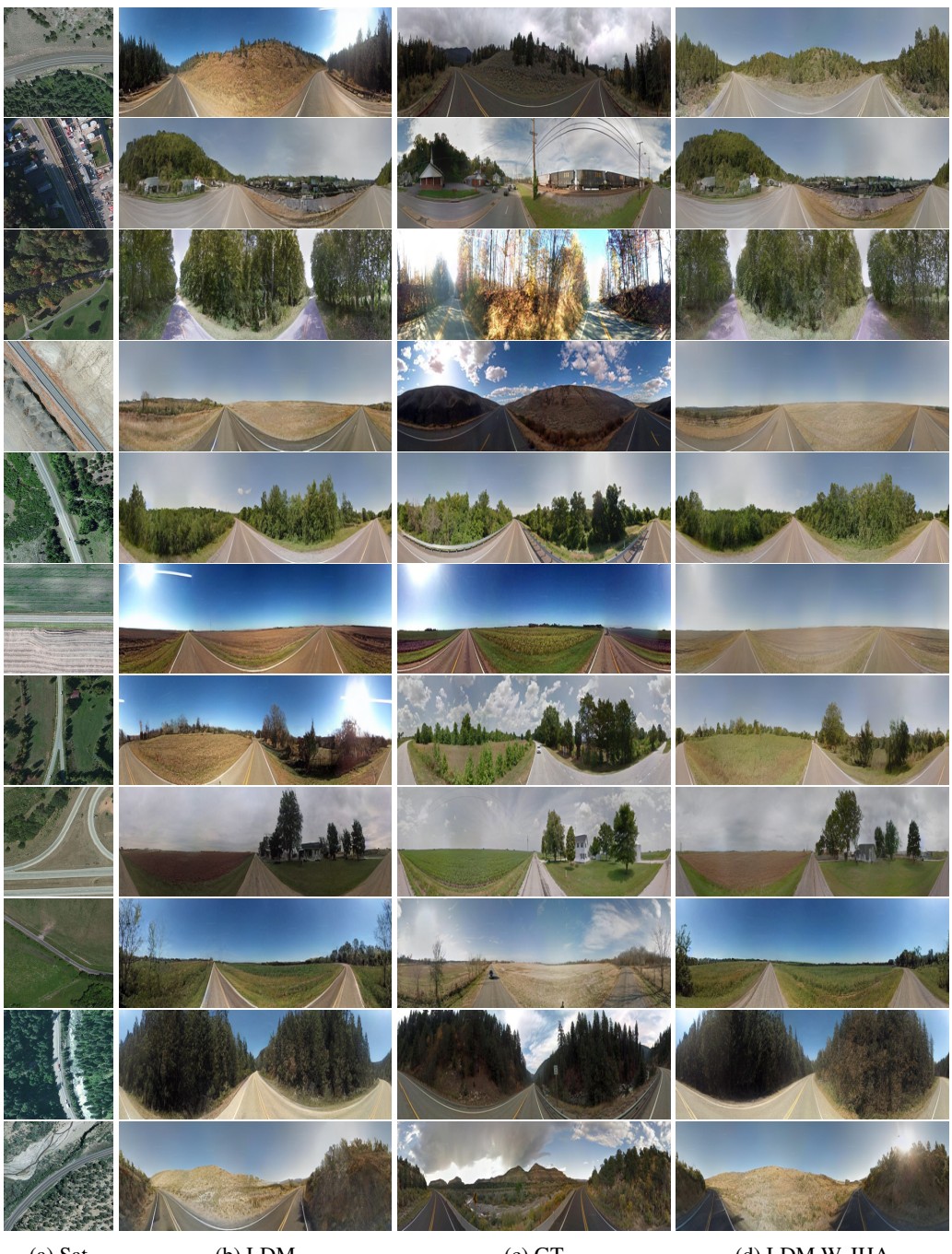

(a) Sat                    (b) LDM                    (c) GT                    (d) LDM W. IHA

Figure 17: Results of IHA on the CVUSA dataset.

