# OpenReview forum: "Controllable Satellite-to-Street-View Synthesis with Precise Pose Alignment and Zero-Shot Environmental Control"
_ICLR.cc/2025/Conference — ICLR 2025 Poster_

### Official Review · Reviewer_KCsS · 2024-10-30

**Soundness:** 2
**Presentation:** 1
**Contribution:** 2
**Rating:** 5
**Confidence:** 5

**Summary:**

Overall, this paper propose a method for satellite-to-street-view synthesis. The authors propose “CROSS-VIEW CONDITIONING MECHANISM” for better geometry consistency, and propose “ITERATIVE POSE ALIGNMENT DURING INFERENCE” at inference stage to further improve the results by utilizing the gradient values between the projected generated image and satellite image. Besides, “TEXT-GUIDED ZERO-SHOT ENVIRONMENTAL CONTROL” utilized the gradient values of CLIP alignment loss to enhance the text controlling.

**Strengths:**

This paper is well-organized and the proposed approach can improve the geometric consistency and textual controlling ability of generative method.

**Weaknesses:**

1. This article lacks a strong motivation and fails to clearly convey the significance of achieving geometric consistency and more accurate street views from satellite images. It doesn't specify under what circumstances a model with enhanced geometric consistency would be preferred. I believe this paper is quite similar to a series of works such as BEVGen[1] and BEVControl[2], which tell the story of geometric consistency with respect to the data augmentation. The author may benefit from expanding this point.
[1]. BEVGen: Street-View Image Generation from a Bird's-Eye View Layout
[2]. BEVControl: Accurately Controlling Street-view Elements with Multi-perspective Consistency via BEV Sketch Layout

2. Some typos: There is a blank line before almost every line of Equation. They should be deleted. In Line-459, it should be Table 1.

**Questions:**

1. In Equation (4) and (5), could you please provide a comprehensive derivative process? Due to space limitations, I suggest that the detailed process can be placed in the supplementary materials. Besides, it is recommended for the authors to provide more details for the meaning of the derived results. What benefits does this derivation bring to subsequent model design? How do model design and formula derivation relate to each other?

2. In Method, the proposed “CROSS-VIEW CONDITIONING MECHANISM” injects the controlling information into the diffusion model. Actually, it is like to introduce a query to extract the features from the satellite image. However, the authors’ statement makes me a little confused. Firstly, the authors state that \mathbf{A} represents the confidence. \delta{h} represents the height offset, but there is no evidence shown here. If they are indeed what the author says they are, at least some visualization should be displayed or some reasons should be explained. Besides, the detail of sampling operation should be given. Secondly, this module is named Geometric Cross-Attention (GCA), but there is no ‘attention’ mechanism here, equation (7) even does not contain a softmax symbol. Thirdly, what is the benefit of this mechanism brings as compared to the naive cross-attention?

3. In “ITERATIVE POSE ALIGNMENT DURING INFERENCE”, equation (8) and Line-277 also have the sampling operation defined in Line-251, please elaborate it. Why the symbol in equation (8) is different in Line-273?

4. In “TEXT-GUIDED ZERO-SHOT ENVIRONMENTAL CONTROL”, why choosing “\epsilon” to controlling the generative precess with respect	to text prompt? It is weird for “potent satellite image prompts significantly undermine the influence of text prompts during training”. Have you applied the CFG scheme?

**Details Of Ethics Concerns:**

I don't understand why Reviewer 7YpL gives such high rating. It is not feasible to customize a task while forcibly bypassing all associated methods. The approach proposed in this article is entirely a subset of methodologies discussed in works such as BEVGen, BEVControl, and MagicDrive. In addition, this article has a large number of grammatical errors and formula errors (refer to previous comments). I was very surprised to see someone give it 8 points.

---

> ### Author Response · Authors · 2024-11-23
> **Official response by authors to Reviewer KCsS: Part1**
>
> We thank Reviewer-KCsS for the valuable comments and suggestions.
> We have thoroughly revised the paper, improved the writing quality, clarified unclear descriptions, and fixed typos. Please find our detailed response to each question below.
>
> **Q1. Lack of clarity on the significance of achieving geometric consistency and more accurate street views from satellite images**
>
> To address this, we provide a demonstration of how the images generated by our method can enhance downstream tasks, such as ground-to-satellite camera localization, through data augmentation. This analysis is detailed in Appendix (Sec.A.2).
>
> Specifically, we use the recent work by Shi et al. [ECCV2024] as the baseline for ground-to-satellite localization and retrain the model using a combination of the original training images and our generated images as augmented data. The performance comparison is presented in Tab.5 (also included below).
>
> |             |                           | ↓Distance (average) | ↓Distance (median) | ↓Angle (average) | ↓Angle (median) |
> |-------------|---------------------------|--------------------|-------------------|-----------------|----------------|
> | **Test1**   | Wo. augmentation          | 11.11             | 7.646            | 0.1811          | 0.1492         |
> |             | W. augmentation           | **10.88**         | **7.167**        | **0.1810**      | **0.1491**     |
> | **Test2**   | Wo. augmentation          | 14.07             | 10.61            | 0.1570          | **0.1305**     |
> |             | W. augmentation           | **13.94**         | **9.885**        | **0.1569**      | **0.1305**     |
>
>
> The results show that incorporating our generated images leads to improved localization performance, highlighting the practical value of achieving geometric consistency and generating more accurate street views from satellite images. This improvement underscores the significance of our approach in real-world applications.
>
> **Q2. Derivative process of Eq.(4) and Eq.(5) and the interrelationship between model design and formulas.**
>
> As suggested, we present the derivation process in Appendix A.4 and thoroughly revise Sec.3 and Sec.4 to make the correspondence between the formula and model design clearer. Specifically, the proposed geometric cross-attention in Sec.4.1 is incorporated in the framework in a classifier-free manner, corresponding to the formulas in Sec.3.1. Sec.3.2 describes how to further enhance pose alignment and control environmental factors in a classifier-guidance manner, which motivates our model design of the iterative pose alignment in Sec.4.2 and text-guided zero-shot environmental control in Sec.4.3.

---

> ### Author Response · Authors · 2024-11-23
> **Official response by authors to Reviewer KCsS: Part2**
>
> **Q3. Details of Cross-View Conditioning Mechanism**
>
> Apologize for the confusion we made in the initial submission. The mechanism indeed leverages a query to extract the features from the satellite image. We have modified Sec.4.1 to make it clearer, and a brief introduction is below.
>
> Specifically, we assume a set of candidate heights for each pixel in the ground image, and use the height information to find corresponding pixels, $P(u_g, v_g, h_i + \Delta h_i)$ from the satellite image, collectting corresponding features by $V \otimes P(u_g, v_g, h_i + \Delta h_i)$, where $V$ denote the satellite feature maps,  $\otimes$ denotes a sampling operation that extracts features from $V$ according to the projected satellite image coordinates.
>
> The collected satellite feature maps at different heights assumptions are then aggregated via an attention matrix $A_i$ (Eq.(5) in the paper), which has a shape of $HW \times N$, where $H$ and $W$ denotes the height and width of the ground features, and $N$ is the number of different heights utilized. The attention weights $A_i$ are normalized via a softmax function across the $N$ height planes, determining the confidence associated with each plane.
>
>
> **(1) Visualization of the "Cross-View Conditioning Mechanism".**
>
> In Fig.12, Sec.A.6.1 of the Appendix, we provide visualizations of the collected features from satellite images under varying height assumptions, the corresponding weights in the attention matrix $A_i$, and the progressively refined ground feature maps achieved using the GCA mechanism.
>
> In the middle of Fig.12, the top row illustrates the ground feature maps collected from satellite features at different candidate heights $h_i + \Delta h_i$, while the bottom row shows the corresponding attention maps $\\{A_i\\}_{i=1}^N$. During the initial GCA iteration (step 1), the projection plane accurately samples features based on the height assumption, with the attention being evenly distributed across various locations, forming the initial solution. By the second GCA iteration (step 2), the offset $\Delta h_i$ induces positional shifts in the projection plane, as highlighted in the red box. These shifts allow the projection to capture more detailed ground-level features, such as pathways, while also refining representations of other scene elements.
>
> As the GCA iterations progress, the attention maps evolve to focus on salient regions. For planes where $h<0$, attention focuses primarily on the ground, while for planes where $h>0$, it shifts towards higher elements such as trees. This adaptive refinement underscores the ability of GCA to progressively enhance feature representation.
>
>
> **(2) Advantages compared to a simple cross-attention mechanism.**
>
> The proposed GCA module offers the following advantages over a naive cross-attention mechanism.
>
> * Flexibility in Image Generation. By leveraging relative pose between satellite and ground images, the proposed GCA enables the generation of ground images at arbitrary locations on the satellite map. In contrast, the simple cross-attention mechanism cannot handle the relative pose information and requires additional modules to process the relative pose information (e.g., zero123[CVPR2023]). Our method is more flexible and can handle various relative pose differences.
>
> * Avoid redundancy of information. The proposed GCA restricts attention to regions that are geometrically likely to correspond—each ground-view pixel attends only to satellite image regions along its camera ray. This focused attention minimizes noise from irrelevant regions, unlike simple cross-attention, which indiscriminately considers the entire image. This targeted approach improves generation quality, as evidenced by the results in Tab.3, where our method consistently outperforms simple cross-attention.
>
> * Improved computational efficiency and reduced GPU memory usage. GCA achieves significant reductions in computational complexity by employing sparse sampling. For satellite image features of size $S \times S$ and ground image features of size $H \times W$, the complexity of naive cross-attention is $O(S \times S \times H \times W)$. In contrast, our algorithm samples N planes (N=8), with the complexity of sampling being $O(N \times H \times W)$, computing horizontal and vertical coordinate offsets at $O(2N \times H \times W)$, and attention calculation at $O(N \times H \times W)$.  Overall, our complexity is significantly reduced to $O(4 \times N \times H \times W)$ compared to naive cross-attention. In practical experiments, for inference on one example, simple cross-attention consumes 250MB of GPU memory, whereas using GCA reduces this to only 184MB.

---

> ### Author Response · Authors · 2024-11-23
> **Official response by authors to Reviewer KCsS: Part3**
>
> **Q4. Explanation of the sampling operation in Line-251, Line-273, and Line-277.**
>
> The "sample" operation is a process of extracting or selecting certain pixel data from the original image. In PyTorch implementation, we use F.grid_sample for this purpose. The sample operation requires two inputs: the source image and the correspondence of pixel coordinates between the source and target images. In the context of this paper, the source image is represented on the left side of the symbol $\otimes$, while the correspondence is on the right side.
>
> After revision, the sampling operations are now in Line-247, Line-274, and Line-277, respectively. In Line-247, the sample operation involves sampling satellite image features (V) onto corresponding eight height hypotheses, as shown in Fig.12. In Line-274 and Line-277, based on the homography matrix H at that point, the images undergo a homographic transformation, where the correspondence between the source and target images is determined by H. Subsequently, the source image is sampled into the new image, as described in Fig.10. In summary, the symbol $\otimes$ in these three equations all represents the sample operation, with the distinction lying in the correspondence between the source and target images.
>
> **Q5. why choose $\epsilon$ to control the generative process with respect to the text prompt?**
>
> Our primary goal is to leverage gradient information from the CLIP model to ensure that generated samples align closely with the provided textual descriptions. By manipulating $\epsilon$, as outlined in Eq.13, we indirectly adjust $z_{t-1}$ during the diffusion process. This approach effectively integrates the semantic guidance from text prompts while maintaining compatibility with the diffusion framework.
>
> **Q6. Explanation of “potent satellite image prompts significantly undermine the influence of text prompts during training”.**
>
> Thank you for pointing out the ambiguity in our phrasing. Here is a more detailed explanation:
>
> In Tab.4, we evaluated multi-condition generation using Classifier-Free Guidance (CFG) within LDM and ControlNet frameworks. These models used satellite images and environmental descriptions (generated by LLAVA) as dual conditioning inputs. However, we found that text-based control was largely ineffective, and the generated results did not sufficiently reflect the textual descriptions.
>
> The underlying issue stems from the dominance of the satellite image condition during training. Each satellite image corresponds uniquely to a specific ground image, providing rich geometric and texture information. Conversely, environmental descriptions generated by LLAVA often corresponded to multiple possible ground images and lacked precise geometric details, as shown in Fig.13, Sec.A.7 of the Appendix. This discrepancy caused the model to assign significantly more weight to the satellite image condition while deprioritizing the environmental description branch. Over time, this imbalance led to the degradation of the model's ability to utilize text prompts effectively.
>
> In summary, while CFG was employed for multi-condition generation, the stronger and more precise information from satellite images overshadowed the less-detailed text descriptions, undermining the influence of text prompts during training.

---

> ### Author Response · Authors · 2024-11-25
>
> Dear Reviewer KCsS,
>
> We would like to thank you very much for your insightful review, and we hope that our response addresses your previous concerns regarding this paper.
>
> Regarding the issue you are concerned about: in Appendix (Sec. A.2), we have supplemented data augmentation experiments to emphasize the motivation of our paper; in the Appendix (Sec.A.4), we have added formula derivations and based on this, we thoroughly revised Sec.3 and Sec.4 to make the correspondence between the formula and model design clearer; in Sec. 4.1, we have revised and explained the GCA module; in Appendix (Sec. A.6.1), we have included visual experiments of the GCA module; in Appendix (Sec. A.6.2), we have outlined the advantages of GCA; in Fig. 9, Fig. 10, and Fig. 12, we have demonstrated sample operations; in Eq. 13, we have explained the reason for adjusting $\epsilon$ in ZoEC; in Tab. 4 and Appendix (Sec. A.7), we have explained the reasons for the failure of the CFG scheme in our application scenario.
>
> As the discussion period is coming to a close, we kindly invite you to share any additional comments or concerns about our work. We would be more than happy to address them. In the meantime, we hope you might consider revisiting your evaluation and potentially increasing your rating.
>
> Thank you for your thoughtful feedback and consideration! We really appreciate it!
>
> Best regards,
>
> The Authors

---

> > ### Comment · Reviewer_KCsS · 2024-11-26
> >
> > Thanks for the responses.
> >
> > The paragraph contains several typographical errors, such as "w e e mploy" in Line 192, which should be corrected. Additionally, there is a discrepancy between Equation (5) in the previous manuscript and Equation (4) in the revised version; it is unclear which one is incorrect. Furthermore, the statement "which motivates our model design of the iterative pose alignment in Sec. 4.2 and text-guided zero-shot environmental control" lacks a robust justification and requires a more compelling explanation.
> >
> > I appreciate that the authors have included data augmentation studies in their rebuttal. However, the experimental results are not sufficiently rigorous. While I understand the constraints on conducting extensive experiments within a limited timeframe, a comparison with existing methods is essential for meeting the standards of top-tier conferences. Although the proposed idea is not novel to me, I believe that it is acceptable to pursue non-novel research, provided it is well-justified and thoroughly evaluated. So, I cannot improve my rating based on current revision.

---

> > > ### Author Response · Authors · 2024-11-26
> > > **Reply to Reviewer KCsS**
> > >
> > > We sincerely thank Reviewer KCsS very much for the further response.
> > >
> > > **Q1: Different equations in the previous manuscript and revised version?**
> > >
> > > Thank you for pointing out this conflict. We have double-checked the revised version and made sure the equation in the currently revised version is correct. The derivation is provided in Sec.A.4 of the Appendix.
> > >
> > > **Q2: The statement "which motivates our model design of the iterative pose alignment in Sec. 4.2 and text-guided zero-shot environmental control" lacks a robust justification and requires a more compelling explanation.**
> > >
> > > In the equation:
> > >
> > > $z_{t-1} = \mu_{\theta}(z_t, t, c) + \lambda \nabla_{z_{t}} \log p(g_{\text{pose}}|z_{t}) + \gamma \nabla_{z_{t}} \log p(g_{\text{text}}|z_{t})  +  \sigma_t \varepsilon$
> > >
> > > $\mu_{\theta}(z_t, t, c)$ corresponds to our proposed Geometric Cross-Attention module, which represents the process of introducing satellite image conditions into Unet using classifier-free guidance for generation. At this point, we identify two issues:
> > >
> > > (i) implicitly leveraging satellite image features as a condition often leads to pose misalignment in generated street-view images (Fig.6);
> > >
> > > (ii) the lack of sufficient environmental condition annotations and sample imbalance often leads to the failure of adding text conditions (Tab.4).
> > >
> > > To address this issue, we introduce classifier-guidance during the inference stage to enhance the conditions of pose and environmental control. Here:
> > > * The second term, $\nabla_{z_{t}} \log p(g_{\text{pose}}|z_{t})$,  corresponds to the proposed Iterative Pose Alignment, which aims to enhance pose consistency between the noisy latent and the conditioning pose. As illustrated, we compute the gradient of pose difference with respect to $z_t$ and update it accordingly. However, when updating $z_t$ in a pixel-wise manner, the degrees of freedom are too large and are thus hard to optimize. To address this problem, we propose to use a Homography matrix to update $z_t$. Details are provided in Sec.4.2.
> > > * The third term, $\nabla_{z_{t}} \log p(g_{\text{text}}|z_{t})$, corresponds to our proposed Text-guided Zero-shot Environmental Control, which leverages environmental text descriptions to guide the generation process. We leverage a multimodal model to examine the disparity between the generated result during the generation steps and the given environmental description, adjusting $z_{t}$ accordingly. Details are provided in Sec.4.3.
> > >
> > >
> > > Based on this, we can generate ground images that match the given pose and environmental text descriptions.
> > >
> > >
> > > **Q3. Experimental results on data augmentation studies are not sufficiently rigorous.**
> > >
> > > We apologize that we did not really understand this question. We are currently working on demonstrating the performance of other satellite-to-ground generation methods (comparison algorithms) in facilitating cross-view localization performance as data augmentation. The results will be supplied as soon as possible.
> > >
> > > Are these experiments sufficient to address this concern? Could you please let us know if any other demonstrations are required?
> > >
> > >
> > > **Q4. Non-novel research.**
> > >
> > > While our method builds on established concepts such as classifier guidance and classifier-free guidance in diffusion models, we have focused on addressing specific challenges in the satellite-to-street-view synthesis task, introducing contributions aimed at filling critical gaps in this area. Specifically:
> > >
> > > * Ensuring pose alignment between the generated street-view images and the conditioning satellite poses is a significant challenge. Our method, including the proposed geometric cross attention and the iterative Homography alignment, introduces innovative mechanisms to enhance pose consistency, which has been underexplored in previous works on diffusion-based image synthesis.
> > > * The task of generating diverse environmental conditions (e.g., seasons, lighting, weather) in the absence of explicit environmental labels in the training dataset is highly challenging. Our approach proposes a unique text-guided sampling strategy that allows for zero-shot control over environmental attributes, which we believe is a meaningful step forward for controllable generation.
> > >
> > >
> > > These contributions address the practical limitations of prior work and demonstrate the applicability of diffusion models in scenarios that demand both geometric fidelity and flexible conditioning. We hope these points clarify the novelty and significance of our approach.

---

> ### Comment · Reviewer_KCsS · 2024-11-27
>
> How is the generated image in Q1 response comes from? What is the detail of data augmentation strategy? Furthermore, could the authors compare your method with other methods on the performance of data augmentation? Such as BEVGen and BEVControl? Could you conduct data augmentation experiments on KITTI or other autonomous driving datasets?

---

> > ### Author Response · Authors · 2024-11-27
> > **Reply to Reviewer KCsS**
> >
> > We sincerely thank Reviewer KCsS very much for the further response.
> >
> > **Where do the images generated in Q1 come from? Details of the data augmentation strategy.**
> >
> > In this experiment, we use the KITTI dataset [1]. [1] provides ground images captured by vehicles, while [2] collects satellite images for each ground image in the KITTI dataset. In the experiment, we use a ground image resolution of 128x512 and a satellite image resolution of 512x512. For each satellite image in the training set of the KITTI dataset, we use the proposed framework to generate street scene images under various seasonal conditions. During the training of the cross-view localization algorithm, both generated and real data are utilized with a 50% probability each to augment the KITTI dataset. Both with and without data augmentation, the model was trained for 15 epochs. These details have been added to Appendix Sec.A.2.
> >
> > **Compare with other methods on the performance of data augmentation, such as BEVGen and BEVControl**
> >
> > Thank you very much for the suggestion.
> >
> > For the BEVGen and BEVControl, their techniques are developed for different purposes compared to our task. Specifically, they focus on generating multi-view ground views controlled by road lines and foreground objects from the BEV perspective. However, our purpose is to synthesize a single street-view image (e.g., a panorama) purely from a satellite image given a relative pose, without providing information about road lines and foreground objects. Therefore, the contributions proposed by BEVGen and BEVControl, such as multi-view cross attention, foreground and background fusion, and object encoding, are not applicable to our task.
> >
> > However, we understand that the reviewer's intention is for us to include more comparative algorithms to strengthen the persuasiveness of our experiments. Therefore, we choose two representative baselines, Pix2Pix and ControlNet, for the experiment. Pix2Pix is a GAN-based neural network that generates ground images by extracting satellite image information through a U-Net. ControlNet is a diffusion model that generates ground images by using the ground projection image derived from satellite image as a condition. The results are as follows:
> >
> > |                        |                         | Distance average | Distance median | Angle average | Angle median |
> > |------------------------|-------------------------|------------------|-----------------|---------------|--------------|
> > | Test1                       | Wo.gen_data             | 11.1066          | 7.6461          | 0.1811        | 0.1492       |
> > |                        | W.Pix2Pix_gen_data      | 11.7217          | 9.0198          | **0.1810**    | **0.1491**   |
> > |                        | W.ControlNet_gen_data   | 11.1487          | 7.6884          | **0.1810**    | **0.1491**   |
> > |                        | W.Ours_gen_data         | **10.8830**      | **7.1671**      | **0.1810**    | **0.1491**   |
> > | Test2                       | Wo.gen_data             | 14.074           | 10.6127         | 0.1570        | **0.1305**  |
> > |                        | W.Pix2Pix_gen_data      | 14.4443          | 11.2866         | **0.1569**    | **0.1305**   |
> > |                        | W.ControlNet_gen_data   | 14.3064          | 10.9059         | **0.1569**    | **0.1305**   |
> > |                        | W.Ours_gen_data         | **13.9403**      | **9.8854**      | **0.1569**    | **0.1305**   |
> >
> > The experimental results show that compared to other generation methods, combining our proposed generation model can better improve localization performance. The data generated by GAN-based Pix2Pix is very blurry. Adding the generated data from Pix2Pix actually deteriorates the model's perception of details. The data generated by ControlNet, although clear, lacks strong positional constraints. The pose offset leads to incorrect matches in the localization model during training. Compared to other methods, our approach can ensure geometrically consistent generation and text-guided diverse environment creation, thereby achieving the purpose of data augmentation and yielding outstanding results.
> >
> > **Data augmentation experiments on KITTI or other autonomous driving datasets?**
> >
> > Our experiments supplied in Q1 of the initial response and the additional comparisons in this response are both conducted on the KITTI dataset. Apologize for not making this clear in our initial response. We have clarified this and provided technical details for this comparison in the paper (Sec.A.2 of the Appendix). Our source code will also be released to facilitate reproducible research.
> >
> > Please do not hesitate to let us know if you have any further concerns and suggestions. We are devoted to address and incorporate them to improve the quality of this paper.
> >
> > _[1] Vision meets robotics: The kitti dataset_
> >
> > _[2] Beyond cross-view image retrieval: Highly accurate vehicle localization using satellite image_

---

### Official Review · Reviewer_vmd3 · 2024-11-03

**Soundness:** 3
**Presentation:** 3
**Contribution:** 3
**Rating:** 6
**Confidence:** 4

**Summary:**

This paper proposes an innovative approach for synthesizing street-view images from satellite imagery, addressing the challenges of geometric alignment and environmental diversity. The authors introduce a Geometric Cross-Attention mechanism (GCA) and Iterative Homography Adjustment (IHA) to ensure geometric consistency between the generated images and satellite views. Additionally, Zero-shot Environmental Control (ZoEC) is employed to flexibly adjust the environmental conditions of the generated images without the need for additional training data. Experimental results show that this method outperforms existing techniques in both geometric accuracy and environmental diversity, demonstrating potential for applications in urban modeling and virtual reality.

**Strengths:**

1. The Iterative Homography Adjustment mechanism iteratively adjusts the pose of the image during the generation process, significantly improving the geometric consistency between the generated image and the satellite image and ensuring the correct spatial alignment of objects in the scene.
2. The Zero-shot Environmental Control mechanism applies text prompts during inference to control environmental conditions without requiring additional training data. At the same time, ZoEC maintains the spatial structure of the image, ensuring that changes to environmental conditions do not affect the original geometric and semantic consistency of the generated image.
3. Extensive experiments have demonstrated the effectiveness of the proposed methods in the paper.

**Weaknesses:**

1. The English expression is inconsistent, and the organization of the paper needs improvement. Some academic terms are used inconsistently, such as "pose correction" and "pose alignment," which should be uniformly applied throughout the paper.
2. There are some errors in the manuscript's presentation, for example: Figures 1–3 are not properly referenced or explained in the main text; Tables 9–10 lack corresponding table header explanations; and the parameter τ in Equation 9 is not adequately explained, making it difficult to read.
3. As far as I know, the CVUSA dataset contains numerous images taken in rural scenes, with many street-view images that include roads or buildings. The author’s displayed images seem to lack discussion on reconstruction results related to these elements.

**Questions:**

1. The authors should discuss how the Zero-shot Environmental Control (ZoEC) performs under different languages or diverse text descriptions. Can the model maintain environmental adjustment accuracy in multilingual scenarios?

---

> ### Author Response · Authors · 2024-11-23
> **Official response by authors to Reviewer vmd3**
>
> We thank Reviewer-vmd3 for the valuable comments and suggestions. Please find our response below.
>
> **Q1. To refine English expression and standardize academic phrases, such as "pose correction" and "pose alignment".**
>
> We have standardized the terminology to "pose alignment" and made corrections to other terms accordingly.
>
> **Q2. Image citation, caption elucidation, and Explanation of Formulas.**
>
> We have corrected all these mistakes in the main text, marked as blue. The parameter τ in Eq. 9 is a temperature hyperparameter, which has now been revised to Eq. 6.
>
> **Q3: Lack of discussion on reconstruction results related to roads and buildings in the CVUSA dataset.**
>
> We provide additional visualizations of generated results by various methods on complex scenes from the CVUSA dataset in Fig.11, Sec.A.5 of the Appendix. These complex scenes comprise a small portion of the dataset, posing significant challenges for faithful ground-view image recovery.
>
> Among the comparison algorithms, S2S and Sat2Den struggle to capture the unique features of rural scenes, producing blurry and indistinct images. ControlNet shows noticeable geometric misalignments. In contrast, our method, leveraging the proposed Geometric Cross-Attention (GCA) mechanism and Iterative Homography Adjustment (IHA) during inference, demonstrates superior recovery of road structures, closely resembling the ground-truth images.
>
> Buildings, which occupy only a small region in the target ground-view images, present an additional challenge as their facades are not explicitly encoded in the conditioning satellite images. Despite this, our method—guided by the geometric reasoning capabilities of GCA and IHA—recovers buildings in the appropriate regions of the ground-view images and synthesizes plausible facade appearances (as shown in the third example in Fig.11).
>
> As a result, our generated images not only retain strong structural consistency but also exhibit a promising ability to represent buildings and other challenging scene elements.
>
> **Q4. How does ZoEC perform under different languages and textual descriptions? Can it adapt to multilingual scenarios?**
>
> Thank you for this insightful question. The proposed ZoEC framework is capable of handling various languages and textual descriptions, provided there exists a multimodal model capable of mapping these languages to the same latent space as images. In our implementation, we use a frozen CLIP model to handle the language of English and compute losses to guide the generative process. However, our framework is flexible and can accommodate other multimodal models for other languages without requiring additional training. For example, Chinese-CLIP can be integrated to support Chinese text prompts, leveraging its ability to align Chinese text with corresponding image features. We demonstrate this in Fig.14, Sec.A.8 of the Appendix, where the model successfully generates corresponding images following the Chinese language guidance.

---

> ### Author Response · Authors · 2024-11-25
>
> Dear Reviewer vmd3,
>
> We would like to thank you very much for your insightful review, and we hope that our response addresses your previous concerns regarding this paper.
>
> Regarding the issue you are concerned about: We have standardized the use of academic terms, formulas, images, and tables; in Appendix (Sec.A.5), we have supplemented the visual analysis of complex scenes in CVUSA; and in Appendix (Sec.A.8), we have supplemented the performance of ZoEC in multilingual scenarios.
>
> As the discussion period is coming to a close, we kindly invite you to share any additional comments or concerns about our work. We would be more than happy to address them. In the meantime, we hope you might consider revisiting your evaluation and potentially increasing your rating.
>
> Thank you for your thoughtful feedback and consideration! We really appreciate it!
>
> Best regards,
>
> The Authors

---

> ### Comment · Reviewer_vmd3 · 2024-11-26
>
> Thank you for your detailed response. I will revise my rating accordingly.  It is worth noting that, to the best of my knowledge, the citation for your CrossviewDiff is incorrect. I suggest updating it to the correct version.

---

> > ### Author Response · Authors · 2024-11-26
> > **Reply to Reviewer vmd3**
> >
> > We sincerely thank Reviewer vmd3 very much for the further response. We have corrected the citation for CrossviewDiff and updated the correct citation in the latest PDF. Please do not hesitate to reach out if you have any further questions.

---

### Official Review · Reviewer_jsaR · 2024-11-03

**Soundness:** 3
**Presentation:** 2
**Contribution:** 3
**Rating:** 6
**Confidence:** 3

**Summary:**

This paper presents a new method for satellite-to-street view synthesis. This paper mainly achieves this goal by integrating Iterative Homography Adjustment to refine geometric alignment and Text-guided Zero-shot Environmental Control to provide flexible environmental control.

**Strengths:**

1. The proposed method is motivated and effective.

**Weaknesses:**

1. Many important works related to generation are missing in this paper, such as DreamFusion[1], Zero-1-to-3[2],  IM-3D[3] and CAT3D[4]. I strongly suggest that the author reorganize Related Works and make a detailed review and introduction to the development of the generation field.
2. The experimental results do not show notable advantages over previous baselines as reported in Table 1 and Table 2.


[1] DreamFusion: Text-to-3D using 2D diffusion,  Poole et. al.

[2] Zero-1-to-3: Zero-shot One Image to 3D Object, Liu et. al.

[3] IM-3D: Iterative Multiview Diffusion and Reconstruction for High-Quality 3D Generation, Melas-Kyriazi et. al.

[4] CAT3D: Create Anything in 3D with Multi-View Diffusion Models, Gao et. al.

**Questions:**

1. Why choose the affine transformation to describe relative pose shift?
2. Is Clip Score enough it measure text-based generation ability?

---

> ### Author Response · Authors · 2024-11-23
> **Official response by authors to Reviewer jsaR**
>
> We thank Reviewer-jsaR for the valuable comments. Please find our response below.
>
> **Q1. Amendments to the Related Works section are needed to include DreamFusion, Zero-1-to-3, IM-3D, and CAT3D.**
>
> Thank you for pointing out the important works. We have restructured our related work section and included detailed discussions of these papers, marked as blue in Lines 121-124. In short, all these works aim to generate novel view images from given camera poses. The difference is that these works are object-centric, while our work focuses on scene-level and target for challenges where target and source views have significant elevation differences.
>
> **Q2. The improvements in the results displayed in Tab.1 and Tab.2 are not significant.**
>
> Due to the minimal overlap between satellite and ground images, the appearance of generated images might significantly differ from the GT ground images, although the geometric layout might be consistent, making significant improvement difficult.
> To facilitate comprehensive comparisons, we further incorporate semantic consistency between generated and GT images for evaluating different methods' performance, in addition to the evaluation metrics in the initial submission.
> The semantic consistency measures the similarity of deep features extracted by DINO[CVPR2021] and SegmentAnything[CVPR2023]. It can be seen that although the superiority of our method on SSIM, PSNR, and RMSE is insignificant, the perceptual, semantic, and depth consistency achieved by our method is much better than the state-of-the-art approaches.
>
> **Q3. Why choose the affine transformation to describe relative pose shift?**
>
> Thank you for pointing out this important question, and we apologize that this wasn't made clear in our initial submission. When there is only a rotation difference between two cameras, the Homogrphy transform is generally applicable to describe the correspondence relationships between pixels in two images. When there are both rotation and translation differences, the Homography transform can only be used to depict pixel correspondences for scene points on the same plane. In practice, the scene depth exhibits complex variations and thus the pixel correspondences between two images are hard to be described by one single Homography. Thus, we propose to iteratively apply Homography on the noisy latent vectors during the denoising sampling process, and expect that a combination of multiple different Homographies collaborates with the denoising UNet to approximate the complex pixel correspondences caused by relative pose shifts.
>
> We have made this clearer in the paper, marked as blue in Lines 264-267, Appendix (Sec.A.3.1), and have changed the mistakenly used description of "affine" to "Homography".
>
> **Q4. Is Clip Score enough it measure text-based generation ability?**
>
> CLIP has a certain capability to evaluate the similarity between generated images and the given environmental prompts. To obtain more comprehensive results, we supplement the experiments with evaluations using BLIP. We query the image content with the question "What season is it there?" to determine the season depicted in each image. Subsequently, we evaluate the recall rate that aligns with the environmental prompts we provide, as shown in Tab.4. The results demonstrate that our method consistently outperforms other methods across all evaluation metrics.

---

> > ### Comment · Reviewer_jsaR · 2024-11-26
> >
> > Thank you for your detailed response. I believe your rebuttal has adequately addressed my concerns, and I will maintain my original score.

---

> > > ### Author Response · Authors · 2024-11-26
> > > **Reply to Reviewer jsaR**
> > >
> > > We sincerely thank Reviewer jsaR very much for the positive response. We are excited to know that your concerns have been addressed. Your invaluable suggestions greatly enhance the quality of the paper.

---

> ### Author Response · Authors · 2024-11-25
>
> Dear Reviewer jsaR,
>
> We would like to thank you very much for your insightful review, and we hope that our response addresses your previous concerns regarding this paper.
>
> Regarding the issue you are concerned about: in related work, we have included more important works; in Tab1 and Tab2, we have conducted more comprehensive comparisons to demonstrate the advantages of the proposed algorithm; in Line 264-267 and Appendix (Sec.A.3.1), we have explained the reason for choosing the Homography transformation; and in Tab.4, we have supplemented the Blip score to more fully evaluate the text-based generation ability.
>
> As the discussion period is coming to a close, we kindly invite you to share any additional comments or concerns about our work. We would be more than happy to address them. In the meantime, we hope you might consider revisiting your evaluation and potentially increasing your rating.
>
> Thank you for your thoughtful feedback and consideration! We really appreciate it!
>
> Best regards,
>
> The Authors

---

### Official Review · Reviewer_7YpL · 2024-11-06

**Soundness:** 3
**Presentation:** 3
**Contribution:** 3
**Rating:** 8
**Confidence:** 5

**Summary:**

This paper focuses on generating street-view images from satellite imagery while preserving geometric consistency and capturing diverse environmental conditions.

To achieve precise geometric control, the authors propose a novel cross-view attention mechanism paired with an Iterative Homography Adjustment scheme. To address environmental variability, they introduce a Zero-Shot Environmental Control strategy.

Furthermore, the paper introduces new evaluation metrics to assess both semantic and geometric consistency between the generated images and corresponding ground truth images.

Extensive experiments validate the effectiveness of the proposed approach in producing diverse, realistic street-view images.

**Strengths:**

The paper clearly defines the problem by identifying two key challenges in Satellite-to-Street-View Synthesis: geometric consistency and environmental diversity. It effectively addresses these challenges by proposing corresponding solutions.

The innovative Iterative Homography Adjustment scheme stands out as a significant contribution. By iteratively refining the intermediate output based on relative pose differences, this approach corrects pose misalignments. I believe the "iterative adjustment" mechanism not only plays a crucial role in the Satellite-to-Street-View Synthesis task but also holds potential for broader applications in other controllable image synthesis tasks.

The introduction of the Zero-Shot Environmental Control (ZoEC) mechanism is another noteworthy highlight. Compared to existing methods like ControlNet, ZoEC demonstrates superior performance in synthesizing environmental variations through text-based control, enhancing the flexibility and realism of the generated outputs.

The paper is well-written and well-structured, making it easy to follow and understand. The clarity of presentation further strengthens the overall impact of the work.

**Weaknesses:**

Please provide an analysis of how the introduction of the Iterative Homography Adjustment (IHA) affects inference speed. Understanding the trade-off between accuracy and computational efficiency would enhance the practical value of the method.

The paper lacks an ablation study on the proposed Geometric Cross-Attention (GCA) mechanism. Including this analysis would help isolate and highlight the specific contribution of GCA to the overall performance.

While the rationale for introducing a new evaluation metric is understandable, for a fair comparison, it is important to also report results using the evaluation pipeline and metrics from previous works, such as RMSE, PSNR, and SD. This would provide a more comprehensive assessment of the method's performance relative to existing approaches.

The scholarship could be improved by referencing other relevant works in conditional street-view synthesis, such as SCP-Diff: Photo-Realistic Semantic Image Synthesis with Spatial-Categorical Joint Prior [ECCV 2024]. This would situate the proposed method more effectively within the broader research landscape.

**Questions:**

see weakness box.

---

> ### Author Response · Authors · 2024-11-23
> **Official response by authors to Reviewer 7YpL**
>
> We thank Reviewer-7YpL for the valuable comments. Please find our response below.
>
> **Q1. How Iterative Homography Adjustment (IHA) affects inference speed.**
>
> The Iterative Homography Adjustment (IHA) mechanism operates entirely in the latent space, where the feature map resolution is low, rather than mapping it back to image space, avoiding the need for a heavy ground-to-satellite localization network that operates on the original image resolution. As a result, the computational overhead introduced by IHA is small. We conducted evaluations with a batch size of 1, and the results are summarized in the table below.
>
> |  | Memory | Time Cost |
> | ------ | ------ | ------ |
> | Wo.IHA | 20126MB | 5.406s |
> | W.IHA | 21022MB | 5.513s |
>
>
> When comparing memory usage, the baseline model without IHA (Wo.IHA) requires 20,126MB, while the inclusion of IHA (W.IHA) increases memory usage slightly to 21,022MB (896MB increase) due to the addition of a lightweight localization network. In terms of time cost, the baseline model (Wo.IHA) requires 5.406 seconds per image, while the model with IHA (W.IHA) increases this slightly to 5.513 seconds per image—an additional cost of only 0.107 seconds per image. We believe that IHA strikes a favorable balance between performance and computational efficiency, offering promising practical value.
>
> We have added corresponding explanations in the Method Section (Line 295-299) and provide the computation comparison in Appendix (Sec.A.3.2).
>
>
>
> **Q2. Ablation study on Geometric Cross-Attention (GCA).**
>
> Thank you for this suggestion. We have incorporated the ablation study results of GCA in Tab.3 of the paper. Similar to the ablation study experiments on IHA, we measure the pose consistency between the generated images and the given pose by leveraging a cross-view localization model.
> The results demonstrate that both the proposed IHA and GCA improve the pose consistency of the generated images.
>
> **Q3. Performance on RMSE, PSNR, and SD.**
>
> We supply the comparisons in terms of RMSE, PSNR, and SD in the paper (Tab.1 & Tab.2). The results demonstrate that our method consistently outperforms previous state-of-the-art across all evaluation metrics on the CVUSA dataset.
>
> For the VIGOR dataset, our approach does not surpass the GAN-based Sat2Density in pixel metrics such as RMSE and PSNR. This discrepancy arises because pixel-based metrics are more tolerant of blurry images. Our approach, based on a diffusion model, aims to recover finer details in images, often leading to larger pixel differences.
> Nevertheless, our method demonstrates clear advantages in structural, perceptual, semantic, and depth consistency metrics, showcasing its ability to generate high-quality results that align more closely with human perception and practical requirements.
>
> **Q4. Referencing other relevant works such as SCP-Diff [ECCV 2024].**
>
> Thank you for recommending this relevant and interesting work. We have discussed it in the related work section (Lines 115-118). The proposed Noise Prior in SCP-Diff is innovative and efficient. We would like to make further exploration along this direction in our task.

---

> ### Author Response · Authors · 2024-11-25
>
> Dear Reviewer 7YpL,
>
> We would like to thank you very much for your insightful review, and we hope that our response addresses your previous concerns regarding this paper.
>
> Regarding the issues you are concerned about: in Line 295-299 and Appendix (Sec.A.3.2), we have supplemented the discussion on the impact of IHA on reasoning speed; in Tab.3, we have added ablation experiments of the GCA mechanism; in Tab.1 and Tab.2, we have included evaluation results of RMSE, PSNR, and SD; and in the related work section, we have discussed the SCP-Diff as you recommended.
>
> As the discussion period is coming to a close, we kindly invite you to share any additional comments or concerns about our work. We would be more than happy to address them. In the meantime, we hope you might consider revisiting your evaluation and potentially increasing your rating.
>
> Thank you for your thoughtful feedback and consideration! We really appreciate it!
>
> Best regards,
>
> The Authors

---

> > ### Comment · Reviewer_7YpL · 2024-11-25
> > **feedback**
> >
> > I generally like this study and vote accept. Although this field is niche for now, I believe in the long run, it may have an impact on mapless driving and driving simulation. But I can only vote 6 not 8, before its impact on large-scale autonomous driving is demonstrated.

---

> > > ### Author Response · Authors · 2024-11-26
> > > **Reply to Reviewer 7YpL**
> > >
> > > Thank you so much for your consistent support!
> > >
> > > During this discussion period, we conducted experiments on KITTI, one of the well-known autonomous driving datasets, to verify whether our method can aid in vehicle localization in autonomous driving scenarios.
> > >
> > > Specifically, we leveraged the recent work by Shi et al. [ECCV2024] as the baseline for ground-to-satellite camera localization and retrained the model using a combination of the original training images and our generated images as augmented data. The performance comparison is detailed in Tab.5 (also provided below).
> > >
> > > |             |                           | ↓Distance (average) | ↓Distance (median) | ↓Angle (average) | ↓Angle (median) |
> > > |-------------|---------------------------|--------------------|-------------------|-----------------|----------------|
> > > | **Test1**   | Wo. augmentation          | 11.11             | 7.646            | 0.1811          | 0.1492         |
> > > |             | W. augmentation           | **10.88**         | **7.167**        | **0.1810**      | **0.1491**     |
> > > | **Test2**   | Wo. augmentation          | 14.07             | 10.61            | 0.1570          | **0.1305**     |
> > > |             | W. augmentation           | **13.94**         | **9.885**        | **0.1569**      | **0.1305**     |
> > >
> > > The results show that incorporating our generated images leads to improved localization performance, demonstrating the usefulness of our approach to autonomous driving tasks.
> > >
> > > Please let us know if you have any other suggestions to improve the quality of this paper. We are very happy to working on it. You are most appreciated!

---

> > > > ### Comment · Reviewer_7YpL · 2024-11-26
> > > > **Good**
> > > >
> > > > I like this new experiment. Although improvements on rotation estimation is marginmal, it is understandable. Please incorporate it into the manuscript. This makes the study a closed-loop data-synthesis-for-perception study, which follows the path (towards embodied AI) that I treasure and believe in.
> > > >
> > > > I have temporarily raised the recommendation to 8. But, the condition is to incorporate this experiment into the manuscript, with clear technical details stating what is 'Test1' and 'Test2' and how to reproduce these results. If this condition is not met, my true recommendation remains as 6.

---

> > > > > ### Author Response · Authors · 2024-11-27
> > > > > **Reply to Reviewer 7YpL**
> > > > >
> > > > > Thank you very much for the positive feedback. We have included this experiment (with more comparisons) and technical details in Appendix Sec.A.2 in the latest PDF. The technical details are also provided below. Our source code will be released to facilitate reproducible research.
> > > > >
> > > > > **Differences between Test1 & Test2:**
> > > > >
> > > > > The cross-view localization work [1] collects satellite images for each ground image in the KITTI dataset and divides the dataset into Training, Test1, and Test2 subsets. The images in Test1 come from the same area as the images in the training set, while the images in Test2 come from different areas to evaluate the algorithm's generalization ability.
> > > > >
> > > > > **Our implementation details:**
> > > > >
> > > > > In our data augmentation experiment, we use a ground image resolution of 128x512 and a satellite image resolution of 512x512. For each satellite image in the training set of the KITTI dataset, we use the proposed framework to generate street scene images under various seasonal conditions. During the training of the cross-view localization algorithm, both generated and real data are utilized with a 50% probability each to augment the KITTI dataset. In Tab.5, we present the localization accuracy with and without the data augmentation. Both of them are trained for 15 epochs. The data augmentation with our generated data significantly improves cross-view localization performance, demonstrating the usefulness of our approach to autonomous driving tasks.
> > > > >
> > > > >
> > > > > Thanks again for your continuous support. Your invaluable suggestions greatly enhance the quality of the paper.
> > > > >
> > > > > _[1] Beyond cross-view image retrieval: Highly accurate vehicle localization using satellite image_

---

### Meta-Review · Area_Chair_ahtv · 2024-12-18

**Metareview:**

The paper presents innovative methods for enhancing the geometric consistency in satellite-to-street-view synthesis. The Iterative Homography Adjustment (IHA) and Geometric Cross-Attention (GCA) mechanisms aim to improve the geometric alignment, while the Zero-shot Environmental Control  allows for environmental adaptability in the synthesized images. The authors' experiments show promising results in urban modeling and virtual reality applications.

The introduction of IHA and GCA is a strong contribution, with reviewers noting their effectiveness in improving geometric consistency. The ZoEC feature is also highlighted for its ability to adjust environmental aspects without extra data. Additionally, the paper is well-organized and the experiments are thorough, providing a clear understanding of the methods' impacts.

The paper suffers from issues with language clarity and presentation inaccuracies. There are also concerns about the novelty of the proposed methods and a lack of discussion on the influence of IHA on inference speed. The paper could benefit from an ablation study for GCA and a more comprehensive review of related generative work.

Despite these concerns, the decision to accept the paper is based on the novel scientific contributions it makes to the field and the authors' successful effort in addressing the critical issues during the rebuttal phase. The clarification of IHA's computational requirements, inclusion of an ablation study for GCA, and improved metric comparisons have significantly strengthened the paper.

**Additional Comments On Reviewer Discussion:**

During the discussion, Reviewer jsaR and Reviewer vmd3 expressed concerns about the inference speed being potentially affected by IHA. Reviewer KCsS pointed out the absence of an ablation study on the GCA mechanism, which is essential for understanding the contribution of each component.

In response, the authors provided detailed explanations and additional data in their rebuttal, emphasizing the minimal computational overhead of IHA and enhancing the clarity and presentation of the Method Section and Appendix. Furthermore, the authors included an ablation study for GCA, and enriched the comparisons with existing metrics.

In reaching the final decision, I considered the extent to which the authors addressed the concerns raised during the review. The authors' thorough rebuttal and significant manuscript improvements have effectively resolved the issues, leading to acceptance.

---

### Decision · Program_Chairs · 2025-01-22

Accept (Poster)